# NEURAL FORCE FIELD: FEW-SHOT LEARNING OF GENERALIZED PHYSICAL REASONING

**Shiqian Li** [1,2,5,6*]   **Ruihong Shen** [3,2,5,6*]   **Yaoyu Tao** [4,1 ✉]   **Chi Zhang** [1,5 ✉]   **Yixin Zhu** [2,1,5,6 ✉]
*Equal contribution   ✉ Corresponding author
[1] Institute for AI, Peking University   [2] School of Psychological and Cognitive Sciences, Peking University
[3] School of EECS, Peking University   [4] School of Integrated Circuits, Peking University
[5] State Key Lab of General AI, Peking University
[6] Beijing Key Laboratory of Behavior and Mental Health, Peking University
https://neuralforcefield.github.io/

## ABSTRACT

Physical reasoning is a remarkable human ability that enables rapid learning and generalization from limited experience. Current AI models, despite extensive training, still struggle to achieve similar generalization, especially in Out-of-distribution (OOD) settings. This limitation stems from their inability to abstract core physical principles from observations. A key challenge is developing representations that can efficiently learn and generalize physical dynamics from minimal data. Here we present Neural Force Field (NFF), a framework extending Neural Ordinary Differential Equation (NODE) to learn complex object interactions through **force field** representations, which can be efficiently integrated through an Ordinary Differential Equation (ODE) solver to predict object trajectories. Unlike existing approaches that rely on discrete latent spaces, NFF captures fundamental physical concepts such as gravity, support, and collision in continuous explicit force fields. Experiments on three challenging abstract physical reasoning tasks demonstrate that NFF, trained with only a few examples, achieves strong generalization to unseen scenarios. This physics-grounded representation enables efficient forward-backward planning and rapid adaptation through interactive refinement. Our work suggests that incorporating physics-inspired representations into learning systems can help bridge the gap between artificial and human physical reasoning capabilities.

## 1 INTRODUCTION

Physical reasoning, the ability to understand and predict how objects interact in the physical world, is fundamental to both human intelligence and artificial systems (Spelke, 2022). This capability underlies crucial applications ranging from robotics to scientific discovery, making it a central challenge in AI research (Lake et al., 2017). One of the remarkable aspects of human cognitive capabilities is the ability to rapidly learn from limited examples (Kim et al., 2020; Jiang et al., 2022; Lake & Baroni, 2023; Zhang et al., 2024), especially evident in intuitive physics (Kubricht et al., 2017; Bear et al., 2021). Humans can quickly abstract core physical principles after observing limited physical phenomena, enabling them to predict complex dynamics and interact with novel environments (Spelke & Kinzler, 2007; Battaglia et al., 2013; Bonawitz et al., 2019; Xu et al., 2021).

In contrast, current AI systems face significant limitations in physical reasoning. Despite being trained on gigantic datasets, these models still struggle to achieve human-level generalization, particularly in Out-of-distribution (OOD) settings (Lake et al., 2017; Zhu et al., 2020). The core issue lies in their tendency to overfit observed trajectories rather than capturing inherent physical principles, severely limiting their ability to compose known knowledge and predict outcomes in novel contexts (Qi et al., 2021; Li et al., 2022; Wu et al., 2023). This stark contrast between human capabilities and current model limitations has motivated the search for new approaches that can learn generalizable physical representations from minimal data.

To bridge this gap, we aim to develop agents with few-shot physical learning abilities that achieve robust generalization across diverse environments. This ambitious goal presents three fundamental challenges: (i) **Diverse Physical Dynamics**: Physical systems exhibit intricate and nonlinear

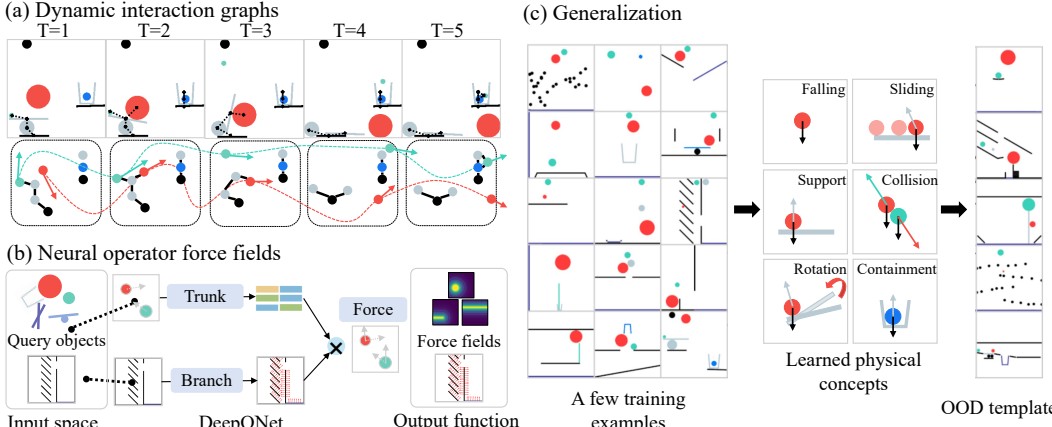

Figure 1: **The framework of NFF.** (a) NFF models complex physical interactions by constructing dynamic interaction graphs from scenes and performing continuous integration on force fields. (b) The force fields are inferred by a neural operator that takes object interactions as input. (c) After learning from a few examples, NFF can capture various physical concepts represented by force fields and generalize to unseen OOD scenarios.

dynamics shaped by complex object properties and interactions. OOD scenarios often present drastically different dynamics from training examples, requiring sophisticated representations that explicitly capture core physical principles. (ii) **Risk of Overfitting**: The few-shot learning setting dramatically increases the challenge of generalization compared to large-scale training approaches. Models must carefully balance between fitting observed examples and extracting broader physical principles. (iii) **Interactive Reasoning**: Effective physical reasoning demands more than passive observation—agents must actively engage with their environment through experimentation and feedback, adapting their understanding based on limited examples.

To address these challenges, we introduce Neural Force Field (NFF), a neural function map from objects to forces acting on continuous Euclidean coordinates for efficient interactive learning and reasoning. At its core, NFF employs a neural operator to learn dynamic force fields from external interventions and object interactions. These predicted forces are then integrated through an ODE solver to compute explicit physical variables such as velocity and displacement, producing interpretable results that align with established physical principles. Please refer to Figure 1 for details.

Our framework offers three advantages. First, by representing physical interactions in low-dimensional neural operator force fields, NFF can rapidly learn fundamental physical concepts from just *a few training examples*. Second, due to the ODE-grounded dynamic graph, NFF can effectively *generalize to OOD scenarios*. Third, the integration of forces through an ODE solver enables *fine-grained physical interactions*, supporting precise modeling of collisions and gravity effects.

We evaluate our method using two state-based physical reasoning benchmarks—I-PHYRE (Li et al., 2023) and N-body problems (Newton, 1833)—as well as one vision-based benchmark, PHYRE (Bakhtin et al., 2019). These tasks feature complex rigid-body dynamics ranging from short-range forces (collision and sliding) to long-range forces (gravity). Our experiments demonstrate that NFF not only learns dynamics efficiently by abstracting physical interactions into force fields but also achieves strong generalization in both within-scenario and cross-scenario settings. Moreover, the framework's physics-based representation enables effective forward and backward planning in goal-directed tasks, consistently outperforming existing methods such as RPIN (Qi et al., 2021), EGNN (Satorras et al., 2021), SEGNO (Liu et al., 2024b), and transformer-based methods (Wu et al., 2023).

## 2 RELATED WORK

**Physical reasoning** Research in physical reasoning has progressed along two main trajectories: passive observation and interactive platforms. The passive observation approach, exemplified by the Violation-of-Expectation (VoE) paradigm (Spelke et al., 1992), evaluates physical understanding by measuring agents' ability to detect violations of physics principles (Baillargeon, 1987; Hespos & Baillargeon, 2001; Dai et al., 2023). While this approach has provided valuable insights into basic physical comprehension, it is limited in assessing active interventions and complex reasoning.

To enable more comprehensive evaluation, interactive platforms such as PHYRE (Bakhtin et al., 2019), the virtual tools game (Allen et al., 2020), and I-PHYRE (Li et al., 2023) have emerged. These environments require agents to actively manipulate objects to achieve specific goals, testing not only prediction capabilities but also planning and reasoning skills. However, current physical reasoning models face two primary challenges: the need for extensive training data (Qi et al., 2021; Li et al., 2023) and limited cross-scenario transferability. While some methods achieve strong performance within specific scenarios (Allen et al., 2020), they often fail to generalize their understanding to novel situations, falling short of human-level reasoning capabilities (Kang et al., 2025).

**Object-centric Dynamic prediction**     The prediction of physical dynamics between objects is fundamental to intuitive physics. Many approaches (Battaglia et al., 2016; Qi et al., 2021) adopt Graph Neural Networks (GNN) as their backbone, capitalizing on their relational inductive bias for modeling object-centric interactions. Key enhancements include integrating E(3) equivariant architectures to generalize across rotations and translations (Satorras et al., 2021; Han et al., 2022), and incorporating NODE for intrinsic continuity modeling to improve long-term prediction accuracy (Poli et al., 2019; Yuan et al., 2024; Liu et al., 2024b). Some works inject physics inductive bias into simulation such as mesh or SDF (Allen et al., 2023; Rubanova et al., 2024) for collision handling and spring-mass models or particle-grid representations (Jiang et al., 2025; Zhang et al., 2025) for deformable objects. However, these methods are mainly designed for particle-based systems which hardly fit into abstract physical reasoning tasks like PHYRE. More importantly, these methods fail to extract robust physical representations, which severely hinders their performance in few-shot learning scenarios. More recent methods like SlotFormer (Wu et al., 2023) and SlotSSM (Jiang et al., 2024) leverage transformer or mamba for temporal sequence modeling with a slot attention (Locatello et al., 2020) module for unsupervised object detection, but still struggle with cross-scenario generalization.

Our work advances continuous-time methods to model complex rigid body interactions via learnable dynamic **force fields**. This approach enables robust cross-scenario generalization while handling non-conservative energy systems. By combining the stability advantages of continuous-time modeling with flexible force field representations, NFF achieves both accurate long-term predictions and strong generalization capabilities without requiring explicit physical priors. More related work can be found in Section A.

## 3 METHOD

### 3.1 THE NEURAL FORCE FIELD (NFF) REPRESENTATION

Traditional approaches to modeling physical interactions typically rely on implicit representations through hidden vectors to describe object interactions. These methods use neural networks to learn state transition functions that map current scene features to future states. However, this purely data-driven approach cannot guarantee physically grounded representations, leading to poor generalization despite intensive training. We introduce NFF, which addresses these limitations by learning physics-grounded, generalizable representations through explicit force field modeling; see Figure 2 for a comparison.

**Predicting force field**     The core of NFF is built on the physical concept of a force field—a vector field that determines the force experienced by any query object based on its state $\mathbf{z}^q$. For a scene with $N$ objects, we model the force field $\mathbf{F}(\cdot)$ at time $t$ using a neural network operating on a relation graph $\mathcal{G}$. The graph contains object nodes $V = \{\mathbf{z}^0(t), \mathbf{z}^1(t), ..., \mathbf{z}^{N-1}(t)\}$. Following neural operator methods (Lu et al., 2021) and graph neural models (Battaglia et al., 2018), we formulate the force field function as:

$$\mathbf{F}(\mathbf{z}^q(t)) = \sum_{i \in \mathcal{G}(q)} \mathbf{W} \left( f_\theta(\mathbf{z}^i(t)) \odot f_\phi(\mathbf{z}^q(t)) \right) + \mathbf{b}, \qquad (1)$$

where $\mathcal{G}(q)$ represents the neighbors of query $q$ determined by contacts or attractions, $f_\theta$ and $f_\phi$ are neural networks with parameters $\theta$ and $\phi$, and $\mathbf{W} \in \mathbb{R}^{d_{\text{hidden}} \times d_{\text{force}}}, \mathbf{b} \in \mathbb{R}^{d_{\text{force}}}$ map hidden features to low-dimensional forces. The state vector $\mathbf{z}$ incorporates geometry states (angle, length, radius), optionally physical attributes (mass), zero-order states (position), and first-order states (velocity, angular velocity). This representation can handle various physical interactions including collision,

sliding, rotation, and gravity, as shown in Figure 1. For image-based NFF, we utilize object masks to replace the geometry states in input.

**Decoding trajectory with ODE** Rather than relying on neural decoders, NFF employs a second-order ODE integrator to compute object trajectories from the learned force field. This approach ensures physically consistent trajectories governed by fundamental principles. The dynamic change of motion for a query state follows:

$$\mathbf{a}^q(t) = \frac{d^2 x^q(t)}{dt^2} = \frac{dv^q(t)}{dt} = \frac{\mathbf{F}(\mathbf{z}^q(t))}{m^q}. \tag{2}$$

While Equation (2) explicitly incorporates object mass $m^q$, our formulation is flexible regarding mass availability. In scenarios with known physical properties, $m^q$ serves as an explicit divisor. Conversely, when mass is not provided as a state input, the network $\mathbf{F}(\cdot)$ learns to implicitly encode mass information into its parameters via end-to-end optimization. In this implicit setting, $\mathbf{F}(\cdot)$ effectively predicts mass-normalized forces (i.e., accelerations) by leveraging correlated features such as object geometry or semantics.

**Solving ODEs** We solve the ODE system using integration methods such as Runge-Kutta and Euler:

$$\mathbf{z}^q(t) = \text{ODESolve}\left(\mathbf{z}^q(0), \mathbf{F}, 0, t\right), \tag{3}$$

$$\begin{cases} \mathbf{x}(t) = \mathbf{x}(0) + \int_0^t \mathbf{v}(t)\, dt, \\ \mathbf{v}(t) = \mathbf{v}(0) + \int_0^t \frac{\mathbf{F}(\mathbf{z}^q(t))}{m^q}\, dt, \end{cases} \tag{4}$$

The NFF framework offers three key advantages over existing approaches. First, it enables few-shot learning of force fields from minimal examples—even single trajectories—through its low-dimensional, physically-grounded representation $\mathbf{F}(\mathbf{z}^q(t))$, adhering to Equation (2). Second, by representing interactions as neural operator force fields, as shown in Equation (1), NFF can generalize the learned force patterns to novel scenarios through simple force summation on new interaction graphs, allowing modeling varying object compositions, interaction types, and time horizons. Third, the high-precision integration decoder, formulated in Equations (3) and (4), enables both fine-grained modeling of continuous-time interactions and efficient optimization for planning tasks through its invertible information flow. This precision supports

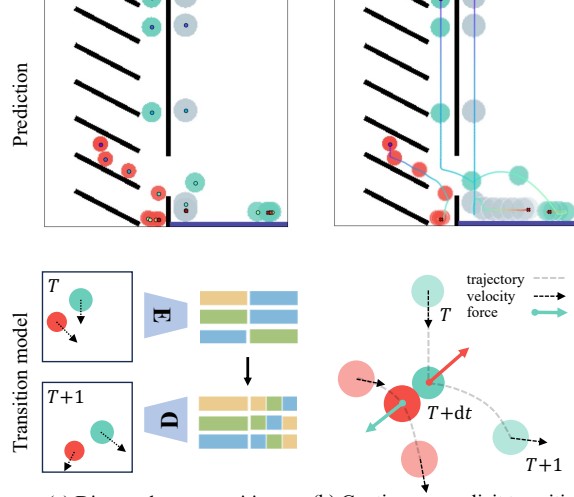

(a) Discrete latent transition    (b) Continuous explicit transition

Figure 2: **Comparison between discrete latent transition in traditional interaction modeling and continuous explicit transition in NFF.** The figures in the first line compare the predicted discrete steps and continuous trajectories, in which the discrete decoding cannot explain how the green ball goes through the black wall. In the second line, the black arrows indicate velocity vectors and the red and green arrows indicate force vectors. Traditional interaction models such as Interaction Network (IN) and SlotFormer decode discrete frames from a learnable decoder. In contrast, NFF decodes trajectories by integrating the continuous force using an ODE solver, which benefits learning detailed interactions such as collisions.

various applications, from determining initial conditions for desired outcomes to designing targeted force fields for specific behaviors. The effectiveness of these three components—ODE grounding, Neural Operator Learning (NOL), and precise integration—is systematically validated through ablation studies in Section 4.5.

**Training NFF** The framework employs autoregressive prediction, using previously predicted states to generate current states. Network optimization minimizes the Mean Squared Error (MSE) loss between predictions and ground truth. To promote learning of local, generalizable dynamics rather than global patterns, we segment long trajectories into smaller units during training. This alleviates the issue of accumulated error in teacher forcing while reducing the complexity in recursive training. In evaluation, the model predicts all future dynamics given only the initial states.

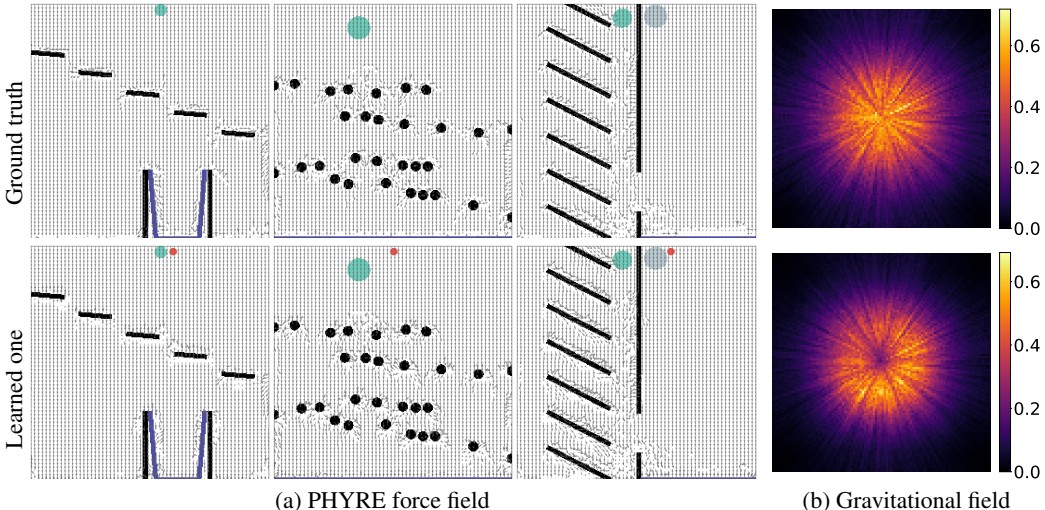

(a) PHYRE force field                    (b) Gravitational field

Figure 3: **Visualization of learned force fields after few-shot learning.** (a) NFF successfully inverts force fields across different templates on PHYRE, consisting of physical behaviors like falling under gravity, sliding with friction, and colliding with momentum transfer. (b) The learned gravitational field distributions in N-body also show accurate force field reconstruction.

## 3.2 INTERACTIVE PLANNING

NFF extends beyond forward prediction to enable mental simulation for planning tasks. Given goal-directed scenarios, NFF can serve as a learned simulator to either search for action sequences that achieve the desired goal state through forward simulation, or optimize initial conditions and system parameters through backward computation.

**Forward planning**    For physical reasoning tasks requiring specific action sequences, we extend the NFF framework to incorporate action effects by including action features in $f_\theta$. Through forward simulation with the NFF model, we sample multiple action sequences $A = \{a_0, ..., a_T\}$ and optimize according to evaluation metrics $R$: $A^\star = \underset{A}{\text{argmax}} R(\mathbf{F}, A)$. The optimal sequence $A^\star$ can then be executed in the physical environment. While the forward simulation may initially deviate from actual observations, NFF's physics-based design enables efficient adaptation through new experimental data, inspired by human trial-and-error learning (Allen et al., 2020). When executed trajectories deviate from the desired goal state, the new state sequences can be directly incorporated into model optimization through the MSE loss, following the same training procedure used in the initial learning.

**Backward planning**    The invertible nature of the NFF formulation makes backward computation particularly efficient. By inverting the time direction of ODE integrator, NFF enables the inversion of initial conditions given a desired goal state: $\mathbf{x}(0) = \mathbf{x}(t) + \int_t^0 \mathbf{v}(t)\, dt$, $\mathbf{v}(0) = \mathbf{v}(t) + \int_t^0 \mathbf{F}(\mathbf{z}^q(t))\, dt$, consistent with the learned physical dynamics encoded in the NFF model, as shown in Section 4.4.

# 4 EXPERIMENTS

## 4.1 ENVIRONMENTS AND DATASETS

We evaluate state-based NFF on I-PHYRE and N-body dynamics and vision-based NFF on PHYRE. Each task is evaluated across three distinct settings: within-scenario prediction, cross-scenario prediction, and planning. Detailed environment and dataset configurations are provided in Section B.

**I-PHYRE**    I-PHYRE (Li et al., 2023) presents a suite of complex physical reasoning puzzles requiring multi-step interventions. The environment incorporates diverse physical interactions including gravity, collision, friction, rotation, spring dynamics, and pendulum motion. It challenges AI agents to solve puzzles with minimal environmental interactions while generalizing to unseen scenarios. For within-scenario prediction, we evaluate on 10 training games that share similar scenarios but require different solutions. The cross-scenario prediction setting extends to 30 novel

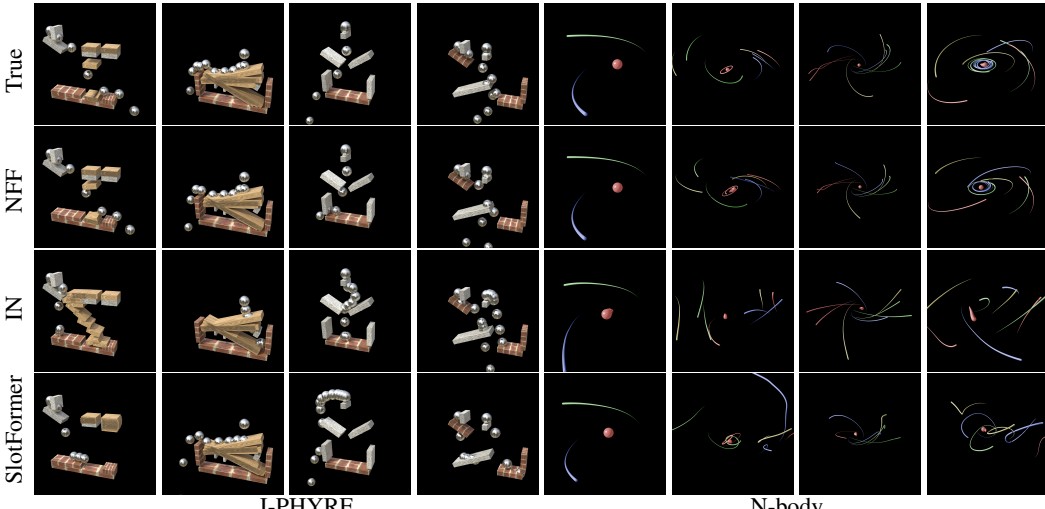

Figure 4: **Trajectory predictions on unseen scenarios after few-shot learning.** After learning from 100 and 200 trajectories on I-PHYRE and N-body systems, respectively, our NFF predictions closely match the ground truth behaviors across diverse scenarios, from rigid body interactions to gravitational dynamics. In contrast, other baselines fail to predict physically plausible dynamics. Additional visualizations are provided in Section H.

games featuring noise, compositional elements, and multi-ball scenarios, with varying object properties such as block lengths, ball sizes, and object positions. In the planning setting, models must generate optimal action sequences to successfully complete each game.

**N-body** The N-body problem (Newton, 1833) tests trajectory prediction for small comets orbiting a massive central planet. Using REBOUND simulator (Rein & Liu, 2012), we generate dynamics by randomly sampling orbital parameters including radii, angles, and masses. This task evaluates the model's ability to infer gravitational laws from limited observations. The within-scenario prediction introduces novel initial conditions and masses, focusing on systems with **1-2 orbiting bodies tracked over 50 timesteps**. Cross-scenario prediction significantly increases complexity by introducing systems with **7 or 9 orbiting bodies tracked over 150 timesteps**. The planning setting challenges models to optimize initial conditions that will evolve to specified target states after 50 timesteps.

**PHYRE** Building on the previous two datasets that used state-based representations of objects, we explore extending NFF to a vision-based physical reasoning benchmark, PHYRE (Bakhtin et al., 2019). Designed for goal-oriented physical reasoning, PHYRE features 25 task templates involving interactive physics puzzles, all solvable by strategically placing a red ball. Each template generates 100 variations of a core task, supporting two distinct evaluation protocols: (i) the within-scenario setting, where models train on 80% of the tasks in each template and are tested on the remaining 20%; and (ii) the cross-scenario setting, which evaluates generalization by training on all tasks from 20 templates and testing on tasks from the 5 unseen ones.

## 4.2 LEARNING FORCE FIELDS FROM A FEW EXAMPLES

We qualitatively evaluate NFF's ability to inverse force fields **from limited observations** without direct access to ground-truth forces; training details are provided in Section C. For I-PHYRE, we train NFF on 10 basic games with 10 trajectory samples each. From these 100 trajectories, the model successfully learns fundamental physical concepts through unified force fields. To verify that NFF learns generalizable physical principles, we examine force responses to varied ball-block interactions under controlled conditions. Figure A1 shows our systematic evaluation where we independently vary individual parameters such as position, velocity, and angle while holding others constant, demonstrating NFF's accurate force response modeling. For the N-body system, we train NFF using 200 randomly sampled trajectories from 2-body and 3-body dynamics. As shown in Figure 3b, the model successfully learns to capture the inverse gravitational field, correctly modeling the distance-dependent centripetal forces governing the mutual attraction between bodies. For PHYRE, we train a vision-based NFF on 12,000 trajectories (20 templates × 20 tasks × 30 actions),

Table 1: **NFF outperforms baselines on prediction tasks.** Comparison of baselines on I-PHYRE and N-body tasks. Results are reported with Standard Error of the Mean (SEM); lower is better (↓) for error metrics (RMSE, FPE, PCE), and higher is better (↑) for correlation (R). NFF consistently yields lower errors and higher correlations, especially under cross-scenario generalization. The [0,3T] settings extend prediction from 50 to 150 steps.

| Metric | Model | I-PHYRE | | N-body | | |
|---|---|---|---|---|---|---|
| | | Within | Cross | Within [0,T] | Within [0,3T] | Cross [0,3T] |
| RMSE ↓ | IN | $0.124 \pm 0.007$ | $0.194 \pm 0.004$ | $0.200 \pm 0.015$ | $0.752 \pm 0.026$ | $6.942 \pm 0.235$ |
| | Slotformer | $0.067 \pm 0.006$ | $0.206 \pm 0.005$ | $0.214 \pm 0.018$ | $1.092 \pm 0.043$ | $2.533 \pm 0.068$ |
| | SGENO | $0.203 \pm 0.070$ | $0.314 \pm 0.106$ | $\mathbf{0.079 \pm 0.009}$ | $0.690 \pm 0.144$ | $2.759 \pm 0.076$ |
| | NFF | $\mathbf{0.048 \pm 0.004}$ | $\mathbf{0.131 \pm 0.004}$ | $0.097 \pm 0.009$ | $\mathbf{0.525 \pm 0.026}$ | $\mathbf{1.226 \pm 0.023}$ |
| FPE ↓ | IN | $0.129 \pm 0.010$ | $0.178 \pm 0.005$ | $0.198 \pm 0.014$ | $0.785 \pm 0.027$ | $11.568 \pm 0.492$ |
| | Slotformer | $0.068 \pm 0.007$ | $0.198 \pm 0.005$ | $0.215 \pm 0.017$ | $1.311 \pm 0.064$ | $2.751 \pm 0.060$ |
| | SEGNO | $0.212 \pm 0.082$ | $0.289 \pm 0.109$ | $\mathbf{0.074 \pm 0.009}$ | $1.535 \pm 0.056$ | $2.302 \pm 0.062$ |
| | NFF | $\mathbf{0.042 \pm 0.004}$ | $\mathbf{0.111 \pm 0.004}$ | $0.104 \pm 0.009$ | $\mathbf{0.592 \pm 0.029}$ | $\mathbf{1.418 \pm 0.029}$ |
| PCE ↓ | IN | $0.004 \pm 0.000$ | $0.005 \pm 0.000$ | $0.012 \pm 0.001$ | $0.029 \pm 0.002$ | $0.101 \pm 0.003$ |
| | Slotformer | $0.002 \pm 0.000$ | $0.004 \pm 0.000$ | $0.013 \pm 0.002$ | $0.035 \pm 0.002$ | $0.053 \pm 0.002$ |
| | SEGNO | $0.008 \pm 0.008$ | $0.017 \pm 0.018$ | $\mathbf{0.005 \pm 0.001}$ | $0.022 \pm 0.004$ | $0.063 \pm 0.001$ |
| | NFF | $\mathbf{0.001 \pm 0.000}$ | $\mathbf{0.003 \pm 0.000}$ | $0.007 \pm 0.001$ | $\mathbf{0.021 \pm 0.002}$ | $\mathbf{0.035 \pm 0.002}$ |
| R ↑ | IN | $0.993 \pm 0.001$ | $0.974 \pm 0.001$ | $0.984 \pm 0.002$ | $0.895 \pm 0.009$ | $0.287 \pm 0.015$ |
| | Slotformer | $\mathbf{0.997 \pm 0.001}$ | $0.967 \pm 0.001$ | $0.977 \pm 0.003$ | $0.854 \pm 0.010$ | $0.4384 \pm 0.019$ |
| | SEGNO | $0.977 \pm 0.026$ | $0.928 \pm 0.064$ | $\mathbf{0.996 \pm 0.001}$ | $0.947 \pm 0.010$ | $0.291 \pm 0.013$ |
| | NFF | $\mathbf{0.997 \pm 0.001}$ | $\mathbf{0.980 \pm 0.002}$ | $\mathbf{0.996 \pm 0.001}$ | $\mathbf{0.949 \pm 0.006}$ | $\mathbf{0.803 \pm 0.010}$ |

which is about 267 times less data than prior SOTA (Qi et al., 2021). As shown in Figure 3a, the model accurately captures force fields across diverse templates.

## 4.3 PREDICTION ON UNSEEN SCENARIOS

We evaluate the learned force fields' predictive accuracy in both within- and cross-scenario settings. For I-PHYRE, we test against 20 ground truth trajectories per game, while for N-body, we evaluate using 200 unseen initial conditions. We compare against established state-of-the-art (SOTA) interaction modeling methods: vanilla IN (Battaglia et al., 2016), SEGNO (Liu et al., 2024b), and SlotFormer (Wu et al., 2023). Note that for fair comparison, we directly feed encoded state information into SlotFormer to avoid inaccurate perception caused by slot attention. More baseline performances on N-body dataset can be found in Section D

**Evaluation metrics** We employ multiple complementary metrics to assess prediction quality. Beyond standard Root Mean Squared Error (RMSE), we introduce Final Position Error (FPE) and Position Change Error (PCE) for detailed performance analysis. FPE quantifies terminal position accuracy, while PCE evaluates the model's ability to capture motion dynamics. Additionally, Pearson Correlation Coefficient (R) measures trajectory shape alignment independent of speed variations. This comprehensive metric suite enables thorough evaluation across temporal and spatial dimensions. Detailed definitions are provided in Section C.2.

**Results** As shown in Figure 4, NFF generates physically plausible trajectories that closely match ground truth behavior, even in previously unseen scenarios. Quantitative results in Table 1 demonstrate NFF's superior performance across all metrics for both I-PHYRE and N-body tasks, with particularly strong results in cross-scenario generalization. While other models exhibit overfitting tendencies that limit the cross-scenario performance, NFF's ability to learn generalizable physical principles enables robust prediction across diverse scenarios.

## 4.4 PLANNING ON UNSEEN SCENARIOS

The trained NFF model can generate plans for novel tasks after learning from limited demonstrations. Unlike prediction tasks that evaluate trajectory accuracy, planning tasks require generating action sequences to achieve specific goals.

**I-PHYRE planning** We implement a 5-round interactive learning protocol. NFF acts as a mental simulator to evaluate 500 randomly sampled action candidates (when to eliminate which block in

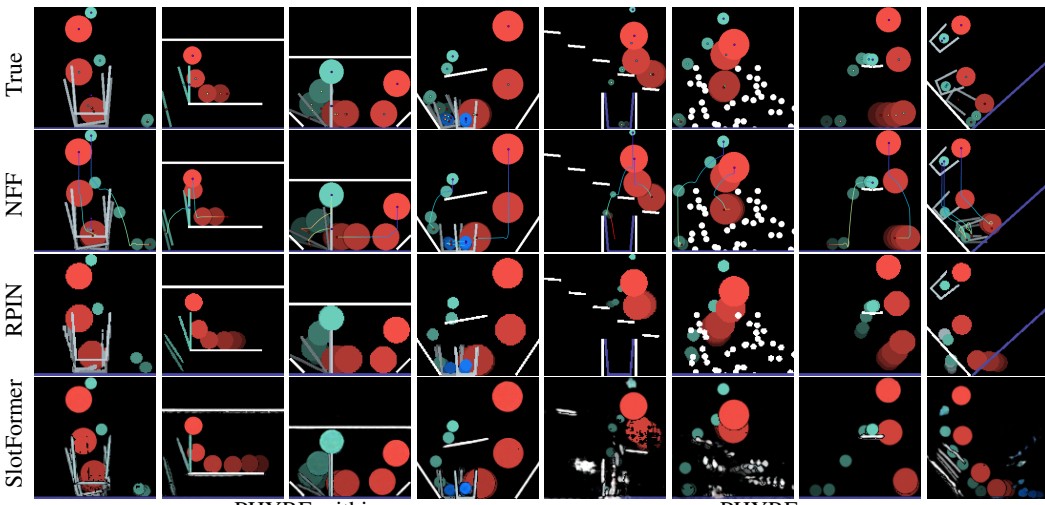

Figure 5: **Trajectory Predictions on PHYRE.** We compare the vision-based NFF, trained on 12,000 trajectories, with SOTA methods trained on 3.2 million trajectories. Prior methods decoding objects from latent spaces often suffer from object inconsistency. For example, in the last column, RPIN mistakenly transforms a gray cup into a gray ball, indicating overfitting to training shapes. SlotFormer also shows object disappearance in cross-scenario tasks. In contrast, NFF produces more accurate predictions while maintaining object consistency.

order to drop the balls into the abyss), selecting the optimal sequence for physical execution. After each execution, the model updates its parameters based on observed outcomes, refining its physics understanding. This updated model then guides subsequent action proposals.

We quantitatively evaluate planning by calculating success probability $p_i$ for each game as the success rate over 20 trials in round $i$, with NFF updating after failures; see detailed results in Section G. Figure 6 compares NFF against random sampling, human performance (from Li et al. (2023)), IN, and SlotFormer using cumulative success probability: $1 - \prod_{i=1}^{n}(1 - p_i)$ for current round $n$. NFF outperforms baselines and approaches human-level performance after refinement (Allen et al., 2020), demonstrating effective few-shot learning. The performance of IN and SlotFormer, falling below random sampling, indicates how inaccurate dynamics modeling compromises planning.

**N-body planning** We focus on determining initial conditions that achieve desired final configurations under celestial dynamics. NFF's inverse simulation enables direct computation of initial conditions through reverse time evolution, while IN and SlotFormer rely on iterative gradient-based refinement. As shown in Table 2, NFF achieves superior planning performance in both within- and cross-scenario settings. Moreover, its reliance on backward integration provides a substantial speed advantage, as further evidenced by the planning time evaluation in Table A3.

**PHYRE planning** In PHYRE, we extend NFF to learn dynamics directly from RGB videos with segmentation masks by applying sparse convolutions to extract geometric features from the masked objects. Following PHYRE's standard evaluation protocol, we first utilize NFF to predict trajectories and then use a pretrained TimeSFormer (Bertasius et al., 2021) as the classifier to determine task suc-

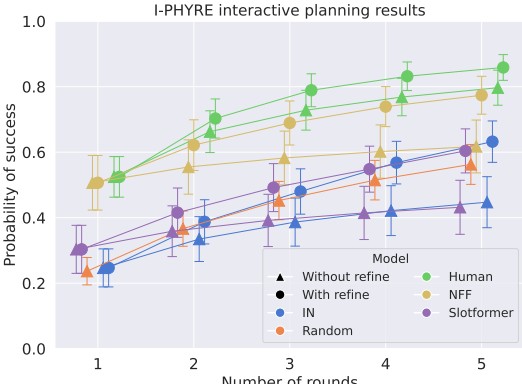

Figure 6: **Interactive planning performance on I-PHYRE.** Comparison of cumulative success probability over 5 planning rounds among human, random sampling, our NFF, IN, and SlotFormer. Error bars show SEM across trials. Our NFF with refinement (Allen et al., 2020) shows continuous improvement across rounds, achieving performance comparable to human (data from Li et al. (2023)), while others show limited performance even with refinement.

cess or failure. Planning performance is measured by the AUCCESS metric (Bakhtin et al., 2019), which captures both effectiveness and efficiency by assigning higher weight to solutions that succeed with fewer attempts. Specifically, each attempt count $k \in 1, 2, 3, \ldots, 100$ is weighted by

$\omega_k = \log(k+1) - \log(k)$, and the final AUCCESS score is computed as: $\frac{\sum_k \omega_k \cdot s_k}{\sum_k \omega_k}$, where $s_k$ is the success rate within $k$ attempts.

Table 2: **N-body system initial condition reconstruction from target configurations.** Results show average MSE ↓ with SEM across trials for both within- and cross-scenario evaluations.

| Scenario | IN | SlotFormer | NFF |
|---|---|---|---|
| Within ↓ | $0.651 \pm 0.021$ | $0.837 \pm 0.029$ | $\mathbf{0.067 \pm 0.010}$ |
| Cross ↓ | $4.654 \pm 0.193$ | $4.018 \pm 0.133$ | $\mathbf{0.140 \pm 0.011}$ |

Table 3: **Planning performance on PHYRE in the cross-scenario setting.** Training sample size (in millions) and AUCCESS ↑ score are reported. Our NFF achieves the best performance with the least data.

| Metric | DQN | RPIN | SlotFormer | NFF |
|---|---|---|---|---|
| Sample (M) | 200.000 | 3.200 | 0.120 | 0.012 |
| AUCCESS | 30.96 | 36.58 | 21.04 | **49.22** |

We compare NFF against the RPIN (Qi et al., 2021) and SlotFormer (Wu et al., 2023). As shown qualitatively in Figure 5, NFF, trained on just 0.012 million trajectories, learns physically grounded dynamics that support effective planning while maintaining higher fidelity and stronger object permanence than RPIN and SlotFormer, particularly in cross-scenario generalization. Quantitative results in Table 3 further demonstrate that NFF outperforms SOTA baselines in the challenging cross-scenario tasks. In contrast, SlotFormer's slot attention leads to limited generalization in unseen objects, resulting in planning failures in cross scenarios; see Figure A6 for more analysis.

## 4.5 ABLATION STUDY

We investigate three key factors that may affect NFF performance, including integration precision, ODE grounding, and NOL. For **integration precision**, we evaluate NFF on the N-body task using Euler integration at two precision levels: $1e-3$, $5e-3$ and adaptive integration with tolerance level: $1e-5$. Table 4 demonstrates that higher integration precision consistently yields better performance. We also explore more integration methods in Section E. To study **ODE grounding**'s impact on generalization, we compare two configurations: our NFF using predicted force fields with ODE integration, and IN using learned state transitions at matching step sizes. Table 4 shows that ODE integration enhances generalization performance. For **NOL**, we replace the DeepONet in NFF with a vanilla MLP. The results in the last column of Table 4 show a consistent increase in generalization errors across settings compared with NFF with DeepONet (column one). These demonstrate that NFF achieves optimal few-shot learning of physical dynamics when combining high integration precision, ODE-based integration, and NOL.

## 5 DISCUSSION

**Extending NFF to real stimuli** While NFF has demonstrated effectiveness in modeling fundamental physical forces like gravity, support, and collision on synthetic datasets, extending NFF to real-world environments remains unexplored. A promising avenue lies in harnessing the strong generalization capabilities of vision foundation models to translate complex, real-world visual inputs into object-centric representations compatible with NFF. Despite the challenges of perceptual noise and diverse physical interactions, we believe this could enable the framework to ground high-level visual perception in explicit force-based dynamics modeling, effectively bridging the gap between video prediction and human-level physical reasoning.

**Limitations** While our experiments focus on generalization across varying object masses and object compositions in controllable abstract reasoning tasks, training a single model with varying friction and elasticity may pose additional challenges. Furthermore, for a clear and constrained validation of learning forces, we assume a deterministic environment with rigid bodies and have not

Table 4: **Ablation results emphasize the need for high integration precision, ODE grounding, and NOL.** We test NFF with different integration precision. The NFF without ODE grounding degenerates to IN with the same integration precision (5e-3) and the NFF without NOL utilize an MLP as force predictor instead of DeepONet. NFF with high integration precision, ODE, and NOL yields the best generalization results.

| Scenario | NFF 1e-3 | NFF 5e-3 | NFF Adaptive | w/o ODE | w/o NOL |
|---|---|---|---|---|---|
| Training | $0.080 \pm 0.009$ | $0.107 \pm 0.013$ | $0.072 \pm 0.008$ | $\mathbf{0.044 \pm 0.005}$ | $0.056 \pm 0.005$ |
| Within | $\mathbf{0.097 \pm 0.009}$ | $0.120 \pm 0.011$ | $0.092 \pm 0.007$ | $0.354 \pm 0.023$ | $0.120 \pm 0.021$ |
| Within [0,3T] | $\mathbf{0.525 \pm 0.026}$ | $0.557 \pm 0.025$ | $0.613 \pm 0.037$ | $0.949 \pm 0.041$ | $0.540 \pm 0.042$ |
| Cross | $\mathbf{1.226 \pm 0.023}$ | $1.251 \pm 0.025$ | $1.788 \pm 0.375$ | $3.518 \pm 0.061$ | $1.347 \pm 0.128$ |

explored the model's performance in stochastic environments or those containing soft bodies and fluids.

**Impact statements**   The proposed NFF framework could enable more sample-efficient and interpretable physical modeling, reducing data needed for training. We acknowledge potential risks if misused for simulating harmful physical scenarios. However, the research focuses on basic physical principles and publicly available phenomena, prioritizing transparent and reproducible science.

## 6   CONCLUSION

We present NFF, a force field-based representation framework for modeling complex physical interactions that exhibits human-like few-shot learning, generalization, and reasoning capabilities. Our experiments on three challenging abstract reasoning datasets with complex multi-object interactions demonstrate NFF's ability to not only learn diverse physical concepts and rules such as gravity, collision, friction, and attraction from limited observations but also generalize to unseen scenarios for both prediction and planning tasks. This initial exploration of the force field may provide new perspectives for developing physical world models through representation learning, potentially bridging the gap between human physical understanding and learning-based methods.

**Reproducibility statement**   To support reproducibility, we provide detailed documentation on the data and environment configurations in Section B, describe the model implementation in Section C, explain the evaluation metrics in Section C.2, and present the ablation studies and additional results in Section D, Section E, Section F, and Section G. Both the datasets and code will be made publicly available.

ACKNOWLEDGMENTS

The authors would like to thank Miss Chen Zhen (BIGAI) for making the nice figures, Prof. Yizhou Wang (Peking University) for the helpful discussion, and Minyang Yu (Peking University) for working on the GPT's results. This work is supported in part by the Brain Science and Brain-like Intelligence Technology - National Science and Technology Major Project (2025ZD0219400), the National Natural Science Foundation of China (62376009), the State Key Lab of General AI at Peking University, the PKU-BingJi Joint Laboratory for Artificial Intelligence, the Wuhan Major Scientific and Technological Special Program (2025060902020304), the Hubei Embodied Intelligence Foundation Model Research and Development Program, and the National Comprehensive Experimental Base for Governance of Intelligent Society, Wuhan East Lake High-Tech Development Zone.

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

## A  MORE RELATED WORKS

**Physics-informed neural dynamics**    Physics-informed methods address the representation learning issues by incorporating physical inductive biases via explicit modeling of state derivatives within dynamical systems (Chen et al., 2018; Zhong et al., 2020; Norcliffe et al., 2020). While they exhibit good adherence to physical laws, current systems typically assume energy-conservative systems with simplified dynamics (Greydanus et al., 2019; Cranmer et al., 2020) or static fields (Kofinas et al., 2023). More importantly, these methods face significant challenges in few-shot learning and struggle with cross-scenario generalization (Chen et al., 2020; Yuan et al., 2024), often requiring specific priors and grammars to achieve meaningful transfer (Xu et al., 2021). Although Physics-informed Neural Networks (PINN) can be precise in modeling physical dynamics, they depend heavily on prior knowledge of the underlying equations (Chu et al., 2022).

**Mental simulation for planning**    Mental simulation is a core mechanism for both human and machine planning. Classic reinforcement learning (RL) agents learn policies through trial-and-error (Mnih et al., 2015; Silver et al., 2016), while model-based RL incorporates explicit rollouts for decision making (Jain et al., 2020). Human studies, such as the Virtual Tool Game, show that people exploit internal physics models to predict imagined outcomes and guide interventions (Allen et al., 2020), inspiring computational work that embeds neural simulators into planning (Parés-Morlans et al., 2025). More recently, generative approaches like diffusion-based planners leverage stochastic simulation to generate diverse, adaptable trajectories in high-dimensional spaces (Janner et al., 2022; Chi et al., 2023), though at the cost of large data requirements. Yet, much prior work still assumes known dynamics or heavy data availability, leaving the few-shot nature of physical reasoning an open challenge.

**Physics-aware video generation**    Recently, video generation models have made significant progress in producing visually realistic scenes, highlighting their potential to serve as "world simulators" by predicting the next frame (Ho et al., 2022; Blattmann et al., 2023). However, research has shown that these models, despite trained on millions of videos, still lack physical commonsense and violate Newtonian laws (Kang et al., 2025; Motamed et al., 2025). To tackle this problem, (Liu et al., 2024a) introduced a physics simulator for producing high-quality physics-consistent videos. However, these methods mainly focus on video synthesis, falling short of physical reasoning in few-shot scenarios.

## B  ENVIRONMENT AND DATASET CONFIGURATIONS

In this section, we describe the environments and datasets used in our experiments, covering three distinct domains: I-PHYRE, N-body simulations, and the PHYRE physical reasoning benchmark. Each environment presents unique challenges and dynamics, requiring tailored data generation and simulation procedures. We outline the structure of the training, within-scenario, and cross-scenario datasets, along with the physical parameters and recorded features essential for model training and evaluation. These configurations ensure both diversity and rigor in testing the generalization capabilities of our models across varying physical setups.

### B.1  I-PHYRE

We adopt the original settings from the I-PHYRE work (Li et al., 2023). The training dataset consists of 10 basic games: support, hinder, direction, hole, fill, seesaw, angle, impulse, pendulum, and spring. For each game, we randomly generate 5 successful and 5 failed action sequences. The within-scenario setting includes an additional 10 successful and 10 failed action sequences for each of the 10 basic games. The cross-scenario setting contains 10 successful and 10 failed action sequences from 30 unseen games, which include 10 noisy games, 10 compositional games, and 10 multi-ball games. All object masses are set to be equal, and small friction and elasticity coefficients are applied. For each game, we record the center positions, lengths, angles, radii, horizontal velocities, vertical velocities, and rotational velocities of each object, as well as the spring pair indices. Each sequence contains up to 12 objects and spans 150 timesteps.

## B.2 N-BODY

We employed the REBOUND engine to simulate N-body dynamics and considered two types of N-body problems for 3D trajectories: the planetary n-body problem and the cometary n-body problem. In the planetary n-body problem, the comets are initialized with Keplerian velocities, while in the cometary n-body problem, the planets are initialized with escape velocities. Initial positions are sampled from the spherical coordinate system and then converted to Cartesian coordinates. For the training and within-scenario datasets, the radii are sampled within the range of 1 to 3, azimuthal angles are sampled from 0 to $2\pi$, and polar angles are sampled from $-\pi/6$ to $\pi/6$. Masses for the orbiting bodies are sampled from 0.05 to 0.1, while the central massive body is assigned a mass sampled from 3 to 5 or from 7 to 9. All sampling is performed using Latin Hypercube Sampling to ensure comprehensive space coverage, even with sparse samples. The training dataset includes 50 two-body planetary trajectories (1 orbiting body), 50 three-body planetary trajectories (2 orbiting bodies), 50 two-body cometary trajectories (1 orbiting body), and 50 three-body cometary trajectories (2 orbiting bodies), spanning across 50 timesteps. The within-scenario dataset consists of the same four trajectory types as the training set but with different initial conditions, with trajectories spanning 150 timesteps. The cross-scenario dataset contains 50 eight-body planetary trajectories, 50 ten-body planetary trajectories, 50 eight-body cometary trajectories, and 50 ten-body cometary trajectories. For these cross-scenario trajectories, initial radii are sampled from 1 to 5, masses of the orbiting bodies are sampled from 0.05 to 0.15, and the central body mass is sampled from 3 to 9. The trajectory length in the cross-scenario data is 150 steps. We record the masses, 3D Cartesian coordinates, and velocities for training.

## B.3 PHYRE

We conduct experiments on the PHYRE-B tier, using the standard split from fold 1—one of the most challenging folds indicated by its low baseline performance. The goal is to make the green ball touch the purple or blue objects by placing a red ball. This physical reasoning benchmark presents a high level of difficulty for several key reasons. First, it features a diverse range of objects—such as balls, bars, sticks, and jars—with varying shapes and uneven mass distributions, adding complexity to interaction dynamics. Second, the solution space is extremely narrow relative to the vast array of possible actions, rendering random attempts largely ineffective. Third, achieving the goal requires navigating a sequence of intricate physical events, including toppling, rotation, collisions, and frictional forces. We train the model using 20 within-template scenarios, each containing 20 tasks, with 30 actions allocated per task. The model is evaluated on the remaining 5 templates, which were not seen during training. For each sample, we simulate dynamic frames at $256 \times 256$ resolution using a stride of 60 and extract object masks for supervision. To ensure stable training, we also record initial linear and angular velocities for each training chunk. To visualize the force field, we place red balls on a $64 \times 64$ grid, run simulations to generate trajectories, and compute second derivatives as field vectors using finite difference methods.

## C TRAINING DETAILS

### C.1 HYPERPARAMETERS

In I-PHYRE, we train the models using a single NVIDIA A100 Tensor Core 80GB GPU. The force field predictor in NFF is based on a DeepONet architecture, consisting of trunk and branch networks, each a 3-layer MLP with a hidden size of 256. The ODE solver uses the Euler method with a step size of 0.005. The 150-step trajectories are divided into 6-step segments, with 6 trajectories per batch. The learning rate starts at 5e-4 and gradually decays to 1e-5 following a cosine annealing schedule. Training occurs over 3000 epochs with a weight decay of 1e-5 for regularization.

The interaction predictor in IN is a 3-layer MLP with a hidden size of 256, while the decoder is also an MLP of the same size. SlotFormer uses 3 transformer encoder layers, each with 4 attention heads and a feedforward network of size 256. Both IN and SlotFormer are trained with a batch size of 50 trajectories, employing the same segmentation technique. All other hyperparameters are kept consistent with those used in NFF.

In N-body, we train the models using a single NVIDIA GeForce RTX 3090 GPU. For all models, the 50-step trajectories are divided into 5-step segments, with 50 trajectories per batch. The learning rate starts at 5e-4 and gradually decays to 1e-7 following a cosine annealing schedule. Training occurs over 5000 epochs with a weight decay of 1e-5 for regularization. All the other hyperparameters are the same as those used in I-PHYRE.

For training the vision-based NFF in PHYRE, we perform feature extraction from $256 \times 256$ object masks using four sparse convolutional layers with channel sizes of 16, 32, 32, and 32. The model architecture includes a DeepONet, composed of a trunk network and a branch network, each with 3 layers and 128 hidden units. DeepONet takes as input a combination of mask, positional, and velocity features, and outputs the corresponding 2D forces and torques acting on each object. Then, we perform a transform and rotation on the object masks to obtain the frames in the next states. The ODE solver uses the Euler method with a step size of 0.025. We adopt a cosine annealing schedule to adjust the learning rate from 3e-4 to 5e-5, with a batch size of 8. Training is conducted for 500 epochs to ensure convergence, with the 18-step trajectories divided into 2-step segments during training.

## C.2 EVALUATION METRICS

In this study, the chosen evaluation metrics include, RMSE, FPE, PCE, and R. Each of these metrics provides valuable insights into different characteristics of the model's predictions, allowing for a comprehensive evaluation.

**Root Mean Squared Error (RMSE)**    The RMSE, the square root of MSE, shares the same characteristics as MSE in terms of penalizing larger errors. However, it provides the error in the same units as the original data, allowing for a more intuitive understanding of how far off the model's predictions are, on average, in the context of physical trajectories:

$$\text{RMSE} = \sqrt{\frac{1}{n} \sum_{t=1}^{n} (\hat{z}_t - z_t)^2}. \tag{A1}$$

**Final Position Error (FPE)**    The FPE specifically measures the discrepancy between the predicted and actual final position of the object at the end of the trajectory. This metric is crucial for goal-driven physical reasoning tasks where the objects are expected to finally move into a specific area. FPE helps ensure that the model is not only capturing the intermediate trajectory but also predicting the final destination with high accuracy:

$$\text{FPE} = |\hat{z}_{\text{final}} - z_{\text{final}}|. \tag{A2}$$

**Position Change Error (PCE)**    The PCE quantifies the error in the predicted change of position over time, which can be interpreted as a measure of the model's accuracy in tracking the object's velocity during its motion:

$$\text{PCE} = |\Delta\hat{z}_t - \Delta z_t|. \tag{A3}$$

**Pearson Correlation Coefficient (R)**    The R measures the linear relationship between the predicted and actual trajectories. R is useful for assessing how well the model captures the overall trend or pattern of the trajectory, even when the absolute errors might vary. A high correlation suggests that the model is effectively tracking the overall movement, while a low correlation might indicate that the model fails to capture the underlying trajectory pattern:

$$R = \frac{\sum_{t=1}^{n} (\hat{z}_t - \bar{\hat{z}})(z_t - \bar{z})}{\sqrt{\sum_{t=1}^{n} (\hat{z}_t - \bar{\hat{z}})^2 \sum_{t=1}^{n} (z_t - \bar{z})^2}}. \tag{A4}$$

## D    MORE BASELINES ON N-BODY DATASETS

We further compare NFF with other advanced GNN-based methods including Graph Convolution Networks (GCN) (Kipf & Welling, 2017), EGNN (Satorras et al., 2021) and GraphODE (Poli et al., 2019; Luo et al., 2023) on N-body datasets. Since most of these methods assume strong priors (e.g., central forces), they cannot be applied to abstract reasoning tasks such as PHYRE or I-PHYRE.

Table A1: **More baselines on N-body datasets**

| Model | Within [0,T] | Within [0,3T] | Cross |
|---|---|---|---|
| NFF | **0.079** ± 0.009 | **0.525** ± 0.026 | **1.226** ± 0.023 |
| GCN | 0.256 ± 0.021 | 0.890 ± 0.034 | 4.343 ± 0.159 |
| EGNN | 0.096 ± 0.009 | 0.718 ± 0.034 | 8.394 ± 0.278 |
| GraphODE | 0.252 ± 0.020 | 0.974 ± 0.039 | 3.920 ± 0.075 |

Table A2: **Results of different kinds of integration methods.** The computational complexity of the model is proportional to the number of average integration steps (relative to Euler $5e - 3$). Among non-adaptive methods, with the increase of computational complexity, the Heun3 method achieves the best performance. For uniformity across tasks and stability during training, we simply selected the Euler method for integration.

| Method | Average steps (relative) | Within | Cross |
|---|---|---|---|
| **Adaptive methods (tolerance)** | | | |
| Adaptive (1e-4) | 0.28 | 0.095 ± 0.008 | 1.712 ± 0.088 |
| Adaptive (1e-5) | 0.36 | **0.092 ± 0.007** | 1.788 ± 0.375 |
| Adaptive (1e-6) | 0.56 | 0.097 ± 0.008 | 1.609 ± 0.104 |
| Adaptive (1e-7) | 1.07 | 0.095 ± 0.007 | 1.556 ± 0.050 |
| **Non-adaptive methods (step size)** | | | |
| Euler (5e-3) | 1.00 | 0.120 ± 0.012 | **1.251 ± 0.025** |
| Midpoint (5e-3) | 2.00 | 0.097 ± 0.009 | 1.736 ± 0.091 |
| Heun3 (5e-3) | 3.00 | 0.096 ± 0.008 | 1.489 ± 0.050 |
| RK4 (5e-3) | 4.00 | 0.094 ± 0.008 | 1.292 ± 0.038 |

# E    ABLATION STUDIES ON INTEGRATION METHODS

In Table A2, we present additional ablation studies comparing various integration methods. Specifically, we evaluate the performance of different integration orders on the N-body task, including Euler, Midpoint, Heun3, RK4, and adaptive methods. Computational complexity is measured by the average number of integration steps. The results indicate that the Euler method excels in cross-scenario generalization. Despite this, Euler integration has lower computational complexity than the higher-order methods, while maintaining better training stability compared to adaptive methods. These findings suggest that Euler integration is sufficient for our physical reasoning tasks.

# F    PLANNING TIME ON N-BODY

We demonstrate how backward integration accelerates planning in N-body tasks. As shown in Table A3, backward planning is up to 10 times faster than traditional optimization-based planning methods.

Table A3: **NFF enables fast planning through backward integration.** NFF performs planning approximately 10 times faster than optimization-based methods in the N-body task. The value represents the time in seconds required to plan 200 scenarios in parallel, with each experiment conducted three times.

| Model | Type | Within | Cross |
|---|---|---|---|
| IN | Optimization | 13.886 ± 0.225 | 11.693 ± 0.153 |
| SlotFormer | Optimization | 22.098 ± 0.156 | 22.489 ± 0.068 |
| NFF | Backward integration | **1.717 ± 0.023** | **1.716 ± 0.013** |

# G    DETAILED PLANNING RESULTS

We present the detailed planning results of different models across trials in Table A4.

Besides those dynamic prediction models, we also benchmark the planning performance of the large language model GPT-4o with the refinement mechanism. The core idea is utilizing In-Context Learning (ICL) abilities and high-level reasoning abilities of large language models to find correct solutions.

The prompt in Round 1 consists of 3 parts: (1) I-PHYRE description and an elaborately explained example. (2) Other training examples. (3) Description of the testing game setting. In Part 2, we provide GPT-4o with 1 successful solution and 1 wrong solution together with resulting trajectories

of objects for each of the 10 basic games. In the refinement stage (*i.e.*, Round 2 and Round 3), we run the action sequence proposed by GPT-4o on the simulator, Then we offer GPT-4o the resulting trajectory and ask it to modify the previous solution or design a new one.

We test the performance of GPT-4o for both within-scenario planning and cross-scenario planning. Detailed results are presented in Table A5. Within-scenario games contains the same 10 games used in the prompt in Round 1. So the language model can memorize corresponding solutions provided in the prompt to achieve good performance. Cross-scenario games contains the other 30 game settings, on which the performance of GPT-4o is better than other baselines (Random, IN, SlotFormer) in Table A4. The improvement of performance through refinement is witnessed in both within-scenario and cross-scenario games.

The prompt used in Round 1 is shown below.

Table A4: **Planning results in I-PHYRE.** Average probability of succeeding cross-scenario games after $n$ trails from IN, SlotFormer, NFF, human, w/o refining. The gray line indicates the best results among all the AI methods.

| Method | Refine | Round 1 | Round 2 | Round 3 | Round 4 | Round 5 |
|--------|--------|---------|---------|---------|---------|---------|
| Random | - | $0.24 \pm 0.04$ | $0.37 \pm 0.05$ | $0.45 \pm 0.06$ | $0.51 \pm 0.06$ | $0.56 \pm 0.06$ |
| IN | ✗ | $0.25 \pm 0.06$ | $0.33 \pm 0.07$ | $0.39 \pm 0.07$ | $0.42 \pm 0.08$ | $0.45 \pm 0.08$ |
| IN | ✓ | $0.25 \pm 0.06$ | $0.39 \pm 0.07$ | $0.48 \pm 0.07$ | $0.57 \pm 0.07$ | $0.63 \pm 0.06$ |
| SlotFormer | ✗ | $0.30 \pm 0.07$ | $0.36 \pm 0.08$ | $0.39 \pm 0.08$ | $0.41 \pm 0.08$ | $0.43 \pm 0.08$ |
| SlotFormer | ✓ | $0.30 \pm 0.07$ | $0.42 \pm 0.08$ | $0.49 \pm 0.07$ | $0.55 \pm 0.07$ | $0.60 \pm 0.07$ |
| NFF | ✗ | $0.51 \pm 0.08$ | $0.55 \pm 0.08$ | $0.58 \pm 0.08$ | $0.60 \pm 0.08$ | $0.62 \pm 0.08$ |
| NFF | ✓ | $0.51 \pm 0.08$ | $0.62 \pm 0.08$ | $0.69 \pm 0.07$ | $0.74 \pm 0.06$ | $0.77 \pm 0.06$ |
| Human | ✗ | $0.52 \pm 0.06$ | $0.66 \pm 0.06$ | $0.73 \pm 0.06$ | $0.77 \pm 0.06$ | $0.80 \pm 0.05$ |
| Human | ✓ | $0.52 \pm 0.06$ | $0.70 \pm 0.06$ | $0.79 \pm 0.05$ | $0.83 \pm 0.04$ | $0.86 \pm 0.04$ |

Table A5: **GPT-4o's planning results in I-PHYRE.** The GPT-4o refines itself after each round of play and shows increasingly better performance.

| Scenario | Round 1 | Round 2 | Round 3 |
|----------|---------|---------|---------|
| Within | $0.74 \pm 0.08$ | $0.78 \pm 0.07$ | $0.82 \pm 0.07$ |
| Cross | $0.35 \pm 0.07$ | $0.45 \pm 0.08$ | $0.51 \pm 0.04$ |

```
/* PART 1 */
    This is a 2D physics simulation environment. There will be one or
 few red balls, and may appear black blocks, grey blocks and blue
blocks. Lengths of blocks are different.

    All these objects are rigid bodies. When we talk about physical
laws, we consider gravity (there is gravity in this environment),
friction (but the friction coefficients are small), collision between
 rigid bodies, rotation of balls and blocks.
    However, grey blocks and black blocks are fixed and stay still to
 their initial positions. They do not apply any laws of physics (even
 collision won't affect them). The red ball and blue blocks are
controlled by physical laws.
    Initially, all objects are at rest (their velocity and angular
velocity are all 0).

    There may exist 2 types of constraints between two objects:
spring or joint. "Spring" constraint means there is a spring linking
2 objects. Springs in all the games have the same rest length,
stiffness and damping rate (You can infer these properties from
trajectories provided below that contain springs). "Joint" constraint
 means there is a rod supporting 2 objects so their distance is fixed
.
    We emphasize that grey and black blocks are not affected by any
constraints. They are fixed to their initial positions.

    Now, we are going to play a game. The goal is to make all the red
 balls fall down to the bottom of the canvas.
    Let's first define a coordinate on the canvas. Let the origin be
the top left corner. The x-axis is horizontal, from left to right.
The y-axis is vertical, from top to bottom. The canvas size is (600,
600), so the coordinate of the lower-right corner is (600, 600).
    The only operation the player can do is to eliminate any grey
blocks at any time they want (however, we limit the whole game in 15
seconds. If after 15s the red ball still doesn't reach the bottom,
the player loses the game). The player cannot eliminate other objects
, including the red ball, blue or black blocks. Note that if the
player eliminates a grey block which is connected to some other
object with a spring or joint, then the spring/joint between them
will disappear.
```

Now, I will show you a game named "fill". The name is actually the key to solve this problem. In this game, a blue block is placed above a grey block. You need to eliminate the grey block so that the blue block falls down to fill the pit on the "ground" (the ground and pit are built by a few black blocks). Otherwise, the red ball will roll into and get trapped in the pit.

Below is the initial setting of the game. I will describe the meaning of each parameter.

{'block': [[[100.0, 420.0], [400.0, 420.0]], [[100.0, 400.0], [250.0, 400.0]], [[350.0, 400.0], [400.0, 400.0]], [[270.0, 200.0], [330.0, 200.0]], [[270.0, 180.0], [330.0, 180.0]], [[100.0, 150.0], [180.0, 200.0]], [[150.0, 100.0], [150.0, 150.0]]], 'ball': [[120.0, 120.0, 20.0]], 'eli': [0, 0, 0, 1, 0, 1, 1, 0], 'dynamic': [0, 0, 0, 0, 1, 0, 0, 1]}

First comes the `block` items. Each item is in the form of `[[left_x, left_y], [right_x, right_y]]`, namely, the coordinates of the left and right corner of the block. Note that each block is generated by moving a circle (radius = 10) centered from the left corner to the right corner.

Next comes the `ball` items. Now we only have 1 ball in this game. This item looks like `[x, y, r]`, namely, the coordinates of the center and the radius of the ball.

Then comes the `eli` item specifying which of these objects are eliminable. In this game, 'eli': [0, 0, 0, 1, 0, 1, 1, 0]. This is an 8-dimensional vector. The first 7 elements describes the 7 blocks (there are 7 elements in the `block` item), and the last element describes the ball. 0 means the object is not eliminable, and 1 means the object is eliminable.

Finally comes the `dynamic` item, which specifies whether the objects are dynamic. In this game, 'dynamic': [0, 0, 0, 0, 1, 0, 0, 1]. The first 7 elements are the 7 blocks, and the last element is the ball. 0 means the object is static (not apply physical laws), and 1 means the object is dynamic (apply physical laws).

You can infer from `eli` and `dynamic` that the 4th, 6th, and 7th blocks are grey, the 5th block is blue. All other blocks are black.

In other game settings, there may exist `spring` or `joint` items. They look like 'spring': [[6, 7]], which means there is only 1 spring connecting the 6th and 7th object (object can be blocks or balls). 'joint': [[6, 7]] means there is only 1 joint connecting the 6th and 7th object.

Here, we also provide an image of the game.

<image here>

Now you are provided with a successful action sequence that can solve the game. The action sequence is as follows:

[[150.0, 125.0, 0.3], [300.0, 200.0, 1.4000000000000004]]

Action sequence is a list of eliminations [pos_x, pos_y, t], where `pos_x` and `pos_y` must be the center of some block (formally, the center is the average of the coordinates of 2 corners: pos_x = 1 / 2 * (left_x + right_x), pos_y = 1 / 2 * (left_y + right_y)), and `t` is the time when the block is eliminated (t should be in 0.1 ~ 15.0). In this example, there are 2 grey blocks to eliminate. [150, 125, 0.3] means eliminate the vertical block (next to the red ball) at t = 0.3s; [300.0, 200.0, 1.4] means eliminate the horizontal block (which supports the blue block from falling down) at t = 1.4s.

In this setting, all positions that can be eliminated are [[300.0, 200.0], [140.0, 175.0], [150.0, 125.0]].

Now you are provided with the trajectory of the red ball and the dynamic (blue) block. Other blocks are either static (the same as in

```
the initial setting) or eliminated at some time. In this case, there
is only one blue block. If there are multiple blue blocks, their
position at some timestamp will be given by the same order as in `dyn
` item. If there are no blue balls, the `dynamic` item will be an
empty list.
    Below is the trajectory.
    t =  0.000s: {'ball': [[120.0, 120.0]], 'dynamic_block': [[270.0,
 180.0, 330.0, 180.0]]}
    t =  0.167s: {'ball': [[120.0, 121.25]], 'dynamic_block':
[[270.0, 180.03, 330.0, 180.03]]}
    t =  0.333s: {'ball': [[120.0, 125.28]], 'dynamic_block':
[[270.0, 180.03, 330.0, 180.03]]}
    ...

/* PART 2 */
    Now we provide you with more examples from different game
settings. We will provide 10 game settings. For each game setting, we
 will provide you with the initial setting (data + image), and 2
action sequence & trajectory (including 1 successful and 1 failed
ones).

    Game 1: name:angle
        Initial setting: {'block': [[[100.0, 400.0], [400.0, 400.0]],
 [[200.0, 350.0], [210.0, 350.0]], [[200.0, 300.0], [210.0, 300.0]],
[[480.0, 400.0], [550.0, 400.0]], [[390.0, 350.0], [400.0, 350.0]],
[[390.0, 300.0], [400.0, 300.0]], [[100.0, 380.0], [100.0, 360.0]],
[[550.0, 380.0], [550.0, 360.0]], [[200.0, 280.0], [400.0, 280.0]]],
'ball': [[300.0, 250.0, 20.0]], 'eli': [0, 1, 1, 0, 1, 1, 0, 0, 0,
0], 'dynamic': [0, 0, 0, 0, 0, 0, 0, 0, 1, 1]}
        Eliminable positions: [[205.0, 350.0], [205.0, 300.0],
[395.0, 350.0], [395.0, 300.0]]
        Game setting image:

    <image here>

        Successful Case 1:
            Action sequence: [[395.0, 300.0, 1.4000000000000004],
[205.0, 350.0, 2.900000000000001], [205.0, 300.0,
3.9000000000000012], [395.0, 350.0, 5.900000000000002]]
            This action sequence leads to success.
            Trajectory:

            t =  0.000s: {'ball': [[300.0, 250.0]], 'dynamic_block':
[[200.0, 280.0, 400.0, 280.0]]}
            t =  0.167s: {'ball': [[300.0, 250.32]], 'dynamic_block':
 [[200.02, 280.3, 400.02, 280.3]]}
            ...

        Failed Case 1:
            Actions: [[395.0, 350.0, 3.000000000000001], [205.0,
300.0, 4.800000000000002], [205.0, 350.0, 6.400000000000002], [395.0,
 300.0, 6.700000000000002]]
            This action sequence leads to failure.
            Trajectory:

            t =  0.000s: {'ball': [[300.0, 250.0]], 'dynamic_block':
[[200.0, 280.0, 400.0, 280.0]]}
            t =  0.167s: {'ball': [[300.0, 250.32]], 'dynamic_block':
 [[200.02, 280.3, 400.02, 280.3]]}
            ...

    ... <more examples>

/* PART 3 */
```

```
    Your goal is to provide a successful action sequence (that makes
the red ball fall to the bottom of the canvas) under a new game
setting. Below is the new game setting.
    Game name: "angle".
        Game setting:
            ...
        Eliminable positions: ...
        Game setting image:

    <image here>

    Your task is to provide a successful action sequence in the same
format as examples given previously. You should make some analysis or
 explanations.
    Now please provide one successful action sequence. At the end of
your response, please give the entire action sequence in the format
of
    "
        Action sequnce: [[pos_1_x, pos_1_y, t_1], [pos_2_x, pos_2_y,
t_2], ...]
    "
    so that we can extract and test your proposed action sequence
easily.
```

The prompt used in GPT-4o.

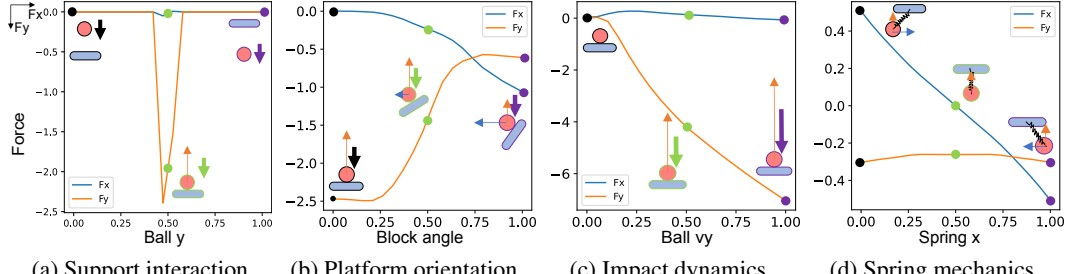

(a) Support interaction    (b) Platform orientation    (c) Impact dynamics    (d) Spring mechanics

Figure A1: **Force response analysis under controlled state variations.** Each plot shows the predicted force components (Fx in blue, Fy in orange) from our trained NFF model, measured in normalized units against different state parameters. The subplots systematically analyze: (a) support forces as a function of ball height, showing characteristic contact response when the ball meets the platform, (b) force decomposition as the platform rotates from horizontal $(0°)$ to vertical $(90°)$, (c) impact forces scaling with the ball's downward velocity, and (d) spring forces varying with horizontal displacement from the equilibrium position. Each plot varies one parameter while keeping all other scene variables constant, demonstrating NFF's learned physical principles, including contact mechanics, geometric reasoning, impact dynamics, and harmonic motion.

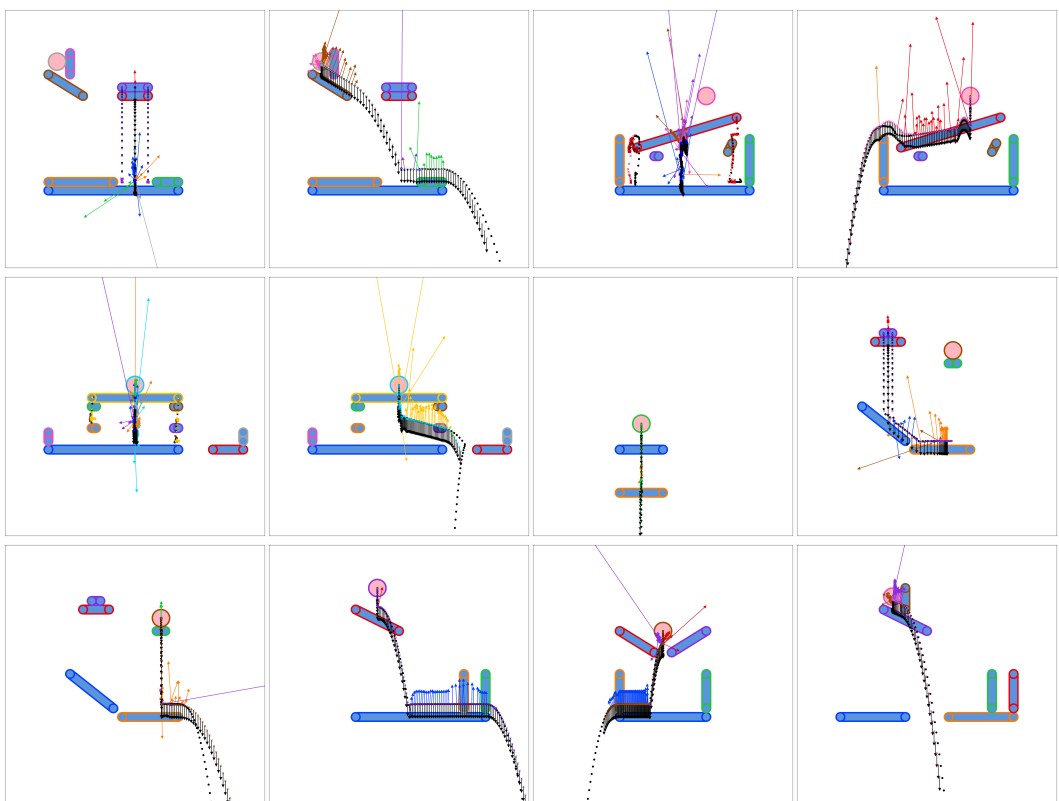

Figure A2: **The inverted forces from NFF and its predicted trajectories of dynamic objects in I-PHYRE within scenarios**. Ground-truth trajectories are represented by black dots, while predicted trajectories are shown with colorful dots. Horizontal and vertical forces are depicted as arrows, while rotational forces are indicated by red or blue dots, signifying negative and positive forces, respectively. The color of the forces indicates which object is exerting the force. Only the initial states are shown for static objects. Some objects will be eliminated during the process. Best seen in videos.

## H   MORE VISUALIZATION

We provide additional visualizations of the learned forces along trajectories in I-PHYRE, as shown in Figure A1, Figure A2, and Figure A3. Further prediction results on the N-body dataset are presented in Figure A4. Additionally, examples of failure cases in NFF predictions are shown in Figure A5. We also visualize the failure encoding of slot-attention in SlotFormer in Figure A6.

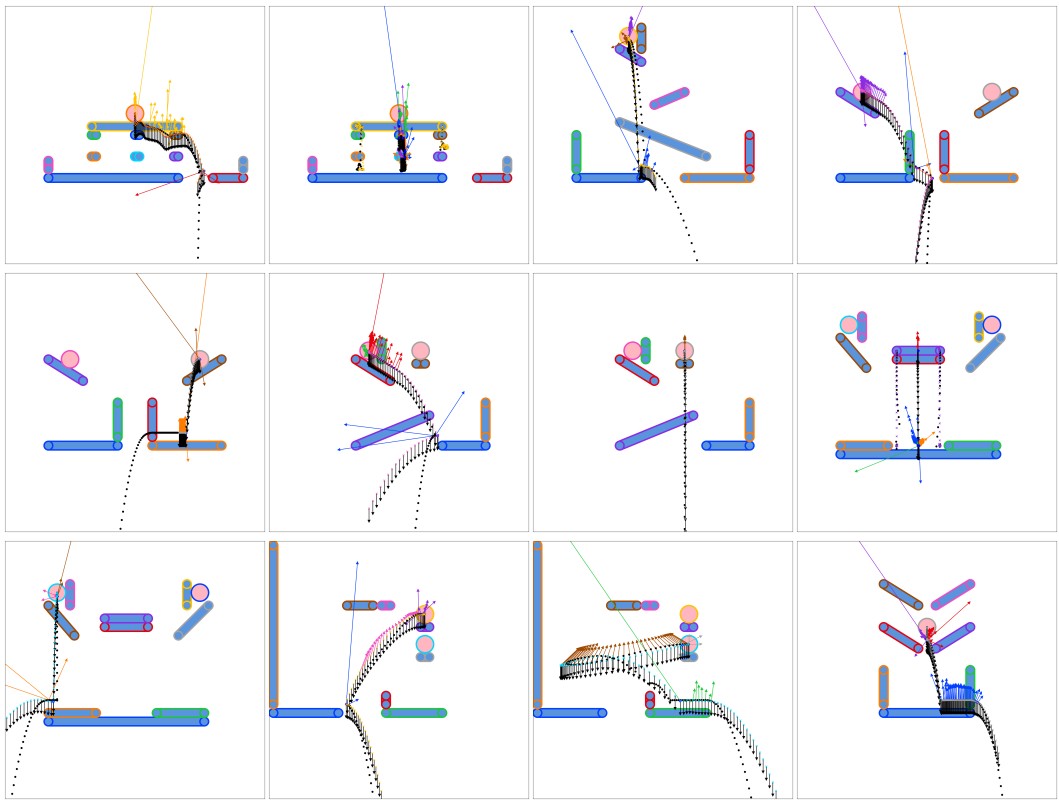

Figure A3: **The inverted forces from NFF and its predicted trajectories of dynamic objects in I-PHYRE cross scenarios**. Ground-truth trajectories are represented by black dots, while predicted trajectories are shown with colorful dots. Horizontal and vertical forces are depicted as arrows, while rotational forces are indicated by red or blue dots, signifying negative and positive forces, respectively. The color of the forces indicates which object is exerting the force. Only the initial states are shown for static objects. Some objects will be eliminated during the process. Best seen in videos.

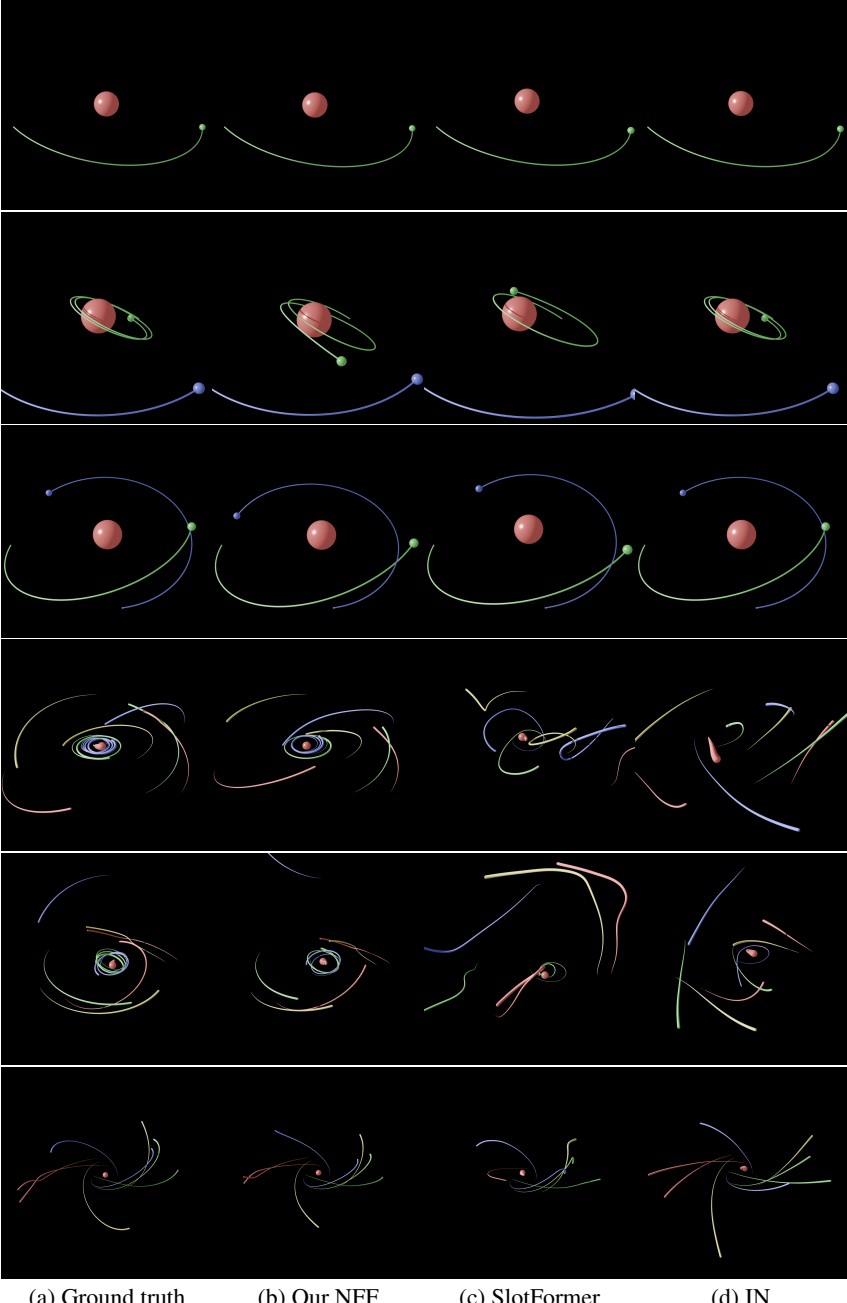

(a) Ground truth     (b) Our NFF     (c) SlotFormer     (d) IN

Figure A4: **Additional prediction results on N-body**. The figure displays predictions from simple N-body dynamics to complex dynamics with more bodies and extended prediction horizons.

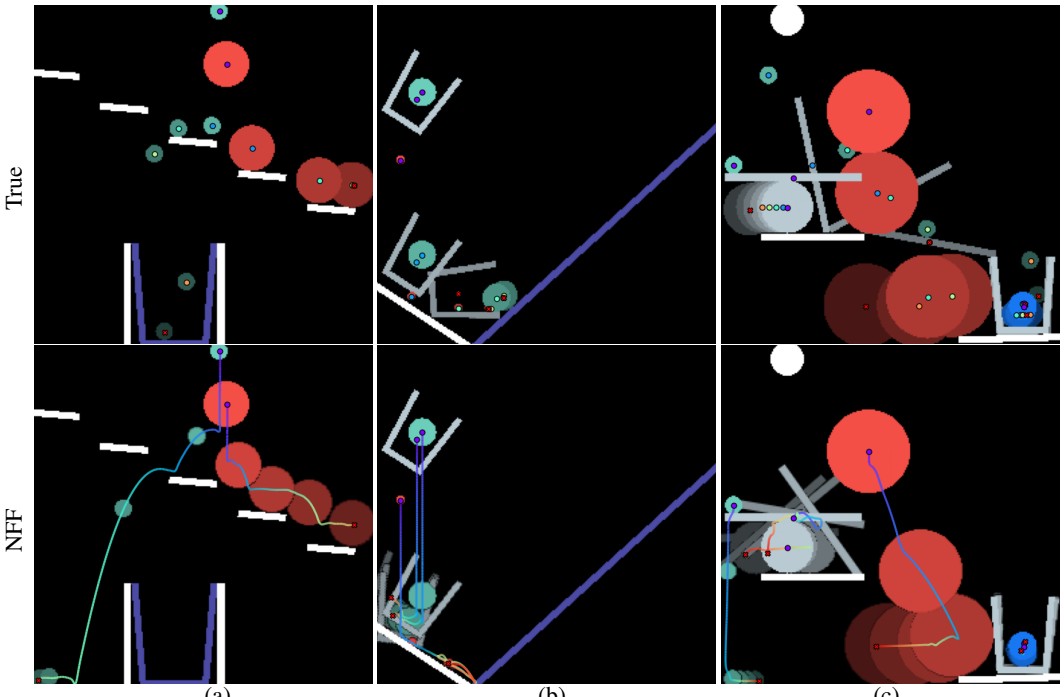

Figure A5: **Examples of prediction failures in PHYRE cross-scenario setting.** (a) Minor prediction inaccuracies accumulate, causing the green ball to fall out of the cup unexpectedly. (b) Extremely tiny red balls introduce unforeseen interactions, potentially resulting in significant dynamic changes, such as cup rotation. (c) Complex interactions between the ball and the seesaw can hinder accurate force prediction.

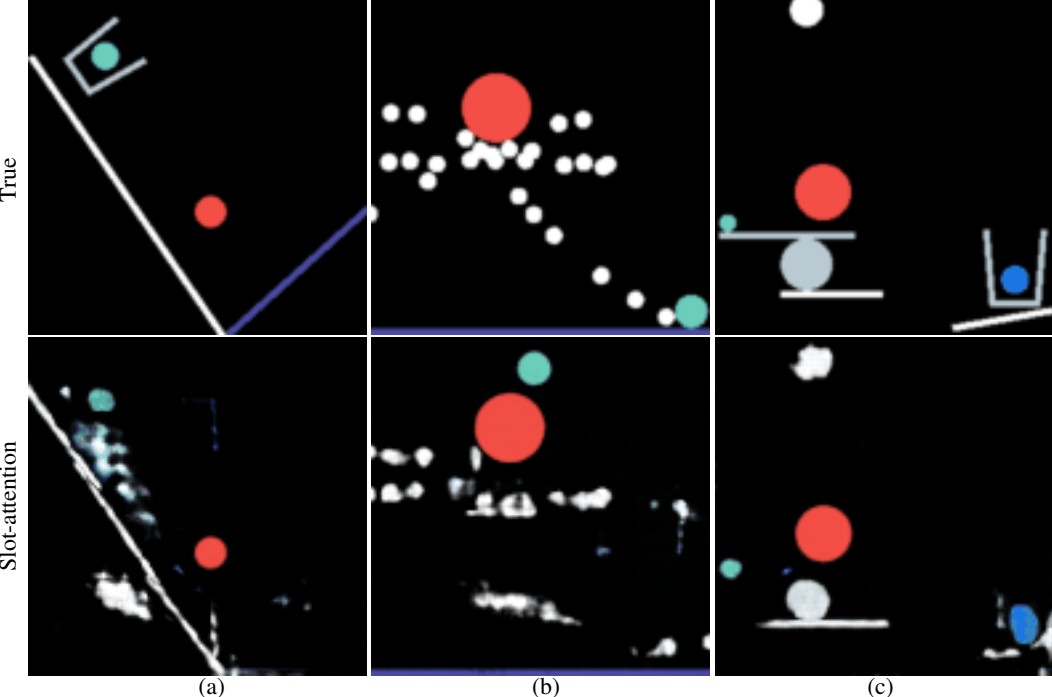

Figure A6: **Examples of failures in Slotformer's pretrained slot-attention module in cross-scenario setting.** The pretrained slot-attention encoder in slotformer, responsible for encoding visual features, exhibits poor generalization to cross-scenario scenes, leading to an inability to capture the precise geometric information crucial for physical reasoning.

