# OpenReview forum: "Neural Force Field: Few-shot Learning of Generalized Physical Reasoning"
_ICLR.cc/2026/Conference — ICLR 2026 Poster_

### Official Review · Reviewer_2R3x · 2025-10-21

**Soundness:** 3
**Presentation:** 4
**Contribution:** 3
**Rating:** 6
**Confidence:** 4

**Summary:**

This paper presents a method for learning physical systems behavior and dynamics by combining an explicit ODE solver working on top of a learned, implicit, force field. The force field is learned as a mapping between a pairs of objects states (position, velocity etc.) in the scene (given the scene graph which denotes which objects interact with which) and the scene space. The training signal comes through prediction error using ground truth trajectories and back-proping through the ODE solver into the neural networks which govern the implied force field.

The method is shown to work nicely on a variety of tasks and an ablation analysis is performed to investigate the effect of different components in the model. Furthermore the model is shown to be useful for physical based reasoning. It is demonstrated that the method is able to learn from significantly

**Strengths:**

Generally speaking I found the paper interesting, well motivated, well executed and enjoyable to read.

* The main contribution of the paper - a combination of an explicit inductive bias in the form of an ODE solver with an implicitly learned force field plays well to the strength of neural networks from one side, and the strength of explicit knowledge on the other side.
* I think this is the first paper I see this specific combination being used.
* The paper is clearly written and enjoyable to read. It is well structured, and well organized. Figures mostly help in understanding and the accompanying video is helpful.
* Experimental validation is satisfactory - covering several tasks, several baselines and ample ablation. Results are convincing, especially on the data efficiency side where one would expect this model to excel at (due to the strong inductive biases).
* While not ground breaking (see below) I think there is a valid contribution to the community here.

**Weaknesses:**

There are several issues, I think, with the paper, some addressable at this time-frame, some probably less so - but none, I think, are critical.

* I think it would be good to show more explicitly how the mapping between the force field acting on each object and scene space - essentially, write down explicitly how to produce the force fields visible in Figure 3. In other words - how is the force field parameterized exactly in terms of coordinates? or in other words, what is d-force and how does it relate to space?

* While there is a nice set of experiments in the paper - they all involve rigid object dynamics. I am wondering if it is possible to apply this method to non-rigid dynamics data - i.e, fluid dynamics, weather systems etc. This would be very compelling.

* On a similar note - all the experiments here assume known ground truth states (either provided explictly, or inferred from masked images). How senstitive is the method to errors in that provided state? Is there a way to learn a mapping, for example, from images to state, by utilizing these inductive biases, or would the system collapse due to inaccuracies?

Given clarifications to the above I will consider raising my score after rebuttal as I like the paper!

**Questions:**

See above.

---

> ### Author Response · Authors · 2025-11-21
> **Response to reviewer 2R3x (Part 1)**
>
> > [W1] I think it would be good to show more explicitly how the mapping between the force field acting on each object and scene space - essentially, write down explicitly how to produce the force fields visible in Figure 3. In other words - how is the force field parameterized exactly in terms of coordinates? or in other words, what is d-force and how does it relate to space?
>
> We agree that explicitly defining the mapping from the continuous scene space to the force field enhances clarity. The Neural Force Field (NFF) parameterizes the force field using a **DeepONet Neural Operator**, which serves as a continuous function approximator over the entire scene domain.
> The key to relating the force to continuous space lies in DeepONet's two-branch architecture:
> 1. **Branch Network (The Force Source)**: This network takes the state information of the neighboring **objects exerting the force** (the acting objects) as input.
> 2. **Trunk Network (The Spatial Query)**: This network takes the state information of the object experiencing the force (the acted-upon object), including its **continuous spatial coordinates** ($x, y$), and other features like velocity, as input.
>
> The DeepONet combines these features to predict the instantaneous force.
> The visualization in Figure 3 is generated by treating the scene as a vector field:
> - We **fix the configuration** of the force-exerting objects.
> - We **query the DeepONet** by setting a hypothetical acted-upon object's position to points on a high-resolution grid over the entire scene space.
> - The output at each grid point is the predicted force vector.
> - **Summation**: For scenes with multiple force-exerting objects, the predicted forces are summed up to yield the resulting force field, which is then visualized as vector plots in Figure 3(a).  For Figure 3(b), to better visualize the N-body gravitational field, we utilized the Line Integral Convolution(LIC) method to visualize the gravitational direction. Additionally, we used a heatmap to visualize the magnitude of the gravitational force experienced by objects.
>
> > [W2] While there is a nice set of experiments in the paper - they all involve rigid object dynamics. I am wondering if it is possible to apply this method to non-rigid dynamics data - i.e, fluid dynamics, weather systems etc. This would be very compelling.
>
> Thank you for your suggestion. We have added a set of experiments including deformable objects in 3D space to demonstrate the adaptability of our method; the same approach may also be applicable to fluids. As for PDE systems such as weather systems, we believe the concept of learning a function field using neural operator learning would be helpful, but this is somewhat out of scope for the intuitive physics problem that we primarily focus on.
>
> For deformable objects in 3D space, we create 3000 3D scenes using MPM in which objects fall and collide with each other. The comparison of prediction results with other baseline models show that our model can better learn the underlying physics and generalize to unseen scenarios. Please see the videos of predicted results at this [link](https://anonymous.4open.science/r/ICLR26_Rebuttal-E7D6).
>
> | Model | Generalization to unseen scenes RMSE ↓ | Generalization to longer timesteps RMSE ↓ |
> | :--- | :--- | :--- |
> | VLM-MPM | 0.306 | 0.328 |
> | GCN | 0.134 | 0.174 |
> | Pointformer | 0.096 | 0.129 |
> | **Our NFF** | **0.082** | **0.107** |

---

> > ### Author Response · Authors · 2025-11-21
> > **Response to reviewer 2R3x (Part 2)**
> >
> > > [W3] On a similar note - all the experiments here assume known ground truth states (either provided explictly, or inferred from masked images). How senstitive is the method to errors in that provided state? Is there a way to learn a mapping, for example, from images to state, by utilizing these inductive biases, or would the system collapse due to inaccuracies?
> >
> > Thank you for pointing this out. We acknowledge that in practice, a completely precise state is hard to acquire. We conducted supplementary experiments. We added varying levels of noise, ranging from 1% to 5%, to the ground truth training state obtained for the N-body system. Experimental results indicate that our model won't collapse under noise introduced in the ground truth state and can still learn a relatively accurate representation of the gravitational field under an imperfect state. In contrast, the SEGNO method , which also uses integration, is highly vulnerable to noise.
> >
> > | Type | Model | w/o Noise | Noise (1%) | Noise (3%) | Noise (5%) |
> > | :--- | :--- | :--- | :--- | :--- | :--- |
> > | Within [0, T] | IN | 0.200 ± 0.015 | 0.174 ± 0.017 | 0.235 ± 0.020 | 0.289 ± 0.022 |
> > | Within [0, T] | NFF (5e-3) | 0.120 ± 0.012 | **0.093 ± 0.012** | **0.102 ± 0.012** | **0.133 ± 0.013** |
> > | Within [0, T] | SEGNO | **0.079 ± 0.009** | 0.109 ± 0.016 | 0.155 ± 0.018 | 0.198 ± 0.020 |
> > | Within [0, T] | SlotFormer | 0.214 ± 0.018 | 0.211 ± 0.023 | 0.277 ± 0.023 | 0.364 ± 0.027 |
> > | Within [0, 3T] | IN | 0.752 ± 0.026 | 0.692 ± 0.036 | 0.782 ± 0.032 | 0.962 ± 0.034 |
> > | Within [0, 3T] | NFF (5e-3) | **0.557 ± 0.025** | **0.556 ± 0.037** | **0.633 ± 0.036** | **0.769 ± 0.045** |
> > | Within [0, 3T] | SEGNO | 0.690 ± 0.144 | 1.105 ± 0.273 | 3.952 ± 1.457 | 4.354 ± 1.765 |
> > | Within [0, 3T] | SlotFormer | 1.092 ± 0.043 | 0.941 ± 0.038 | 1.019 ± 0.042 | 1.261 ± 0.052 |
> > | Cross [0, 3T] | IN | 6.942 ± 0.235 | 3.928 ± 0.168 | 3.049 ± 0.137 | 3.051 ± 0.090 |
> > | Cross [0, 3T] | NFF (5e-3) | **1.251 ± 0.025** | **1.338 ± 0.050** | **1.401 ± 0.035** | **1.512 ± 0.049** |
> > | Cross [0, 3T] | SEGNO | 2.759 ± 0.076 | 4.026 ± 0.169 | 48.042 ± 6.311 | 24.668 ± 5.858 |
> > | Cross [0, 3T] | SlotFormer | 2.533 ± 0.068 | 2.794 ± 0.077 | 2.125 ± 0.060 | 2.297 ± 0.068 |

---

> > ### Comment · Reviewer_2R3x · 2025-11-21
> > **Thank you for the detailed response to all my questions.**
> >
> > The added experiments add a lot of value to the paper and I would encourage the authors to include these in the final manuscript (or the supplementary material).
> > This is an excellent rebuttal and I am happy to raise my score.
> > This is a strong contribution to the community and I recommend this paper be accepted to the conference.

---

> > > ### Author Response · Authors · 2025-11-21
> > >
> > > Thank you very much for your positive evaluation and for recommending our paper for acceptance. We appreciate your encouraging feedback and are glad that the additional experiments strengthened the work. We will incorporate these results into the final manuscript (and supplementary material where appropriate) to ensure the contribution is as clear and complete as possible.

---

### Official Review · Reviewer_SNun · 2025-10-31

**Soundness:** 3
**Presentation:** 3
**Contribution:** 2
**Rating:** 6
**Confidence:** 3

**Summary:**

This paper proposes a pipeline, Neural Force Field (NFF), for predicting force fields using neural networks—as opposed to directly predicting velocities or state changes. From what I understand, the main motivation is to exploit ODE integration methods by learning to predict force fields. This allows for continuous integration at arbitrary resolutions. Experiments on three different datasets show the advantage of using NFFs.

**Strengths:**

The paper is written clearly and quite well-organized. The motivation behind most of the design decisions are explained clearly. Experiments are designed in a way to directly test the claims. Overall, I enjoyed the paper and I think it might draw some attention.

**Weaknesses:**

I didn’t follow the reasoning after Eq. 2. If we do not take mass into account, how can we predict the velocity accurately in general? It’s mentioned that mass is absorbed in force through correlated features like size, but is there any experiment that tests this assertion with counterintuitive sizes? Or, to put it in another term, how do the results change with objects of highly varying densities? Do you think adding a mass prediction would help? This part needs to be discussed and motivated in more detail.

I’m not sure if this is due to the space constraints but, Figures 1 and 2 (especially Fig. 2) are a bit condensed and hard to parse.

I'm not fully familiar with the work in this subfield, but it doesn't strike me as something brand new to do force prediction and combine it with ODE integration. Most of the related work is mentioned as it lacks robustness and sample efficiency, which makes it feel like the main contribution is the few-shot performance rather than the force + ODE combination. If that is the case, then I would wonder what's different in this method that makes it more sample efficient and applicable in few-shot scenarios---I don't think it's clear from the related work section. I tried to search for other methods, and found some, but I couldn't be sure if they are working with the same set of assumptions, and therefore, didn't want to speculate. So, I am relying on my fellow reviewers on this point, and might change my score if they think this is very close to some other work.

**Questions:**

Please also consider my concern in the weaknesses section.

Apart from that, I’m not sure whether training the model with new data should be called interactive reasoning. Could you provide some details on the model update? How many optimization steps are done in each round in Figure 6?

All of the compared methods are doing pixel reconstructions as far as I can tell. While the results support that predicting force fields is a better alternative when compared with pixel reconstruction, I’m not sure if it compares the model itself (i.e., the force field + ODE integration) with, say, SlotFormer, since the predictions are in different lands. I wonder how SlotFormer would perform if it’s at least predicting positions (if not forces). Or is it the case for only PHYRE tasks?

---

> ### Author Response · Authors · 2025-11-21
> **Response to reviewer SNun (Part 1)**
>
> > [W1] is there any experiment that tests this assertion with counterintuitive sizes? Or, to put it in another term, how do the results change with objects of highly varying densities? Do you think adding a mass prediction would help?
>
> Yes, we agree that adding a mass prediction would help solve cases with counterintuitive sizes. We would like to clarify that for the N-Body dataset, we explicitly included mass as known information. However, for PHYRE and I-PHYRE, we followed their original settings in which the object mass cannot be directly obtained. Fortunately, even if we do not know the ground-truth mass, our model can still implicitly encode this from the visual cues (e.g. sizes) into its parameters. In this case, the output $\mathbf{F}$ effectively corresponds to accelerations, or equivalently, forces under the assumption that all object masses are equal to 1. For objects with counterintuitive sizes, their masses cannot be inferred from static states alone, and observed trajectories are required to estimate them. To make this clearer and more general, we have modified the equation to include an explicit mass term (see equation 2 and 4).
>
> > [W2] I’m not sure if this is due to the space constraints but, Figures 1 and 2 (especially Fig. 2) are a bit condensed and hard to parse.
>
> We have revised our manuscript and have taken the following actions to improve clarity of these figures:
> 1. **Detailed captions**: We added more legends to Figure 2 to illustrate the operating mechanism of our method.
> 2. **Modify the layout of the figures**: We enlarged the spaces between subfigures for better clarity.
>
> > [W3] the main contribution is the few-shot performance rather than the force + ODE combination
>
> We appreciate the reviewer's questions regarding the difference between our method and others.
> The core difference that enables this performance is our specific representation: Neural Force Field (NFF), which is formulated using an **explicit force field** trained via a **neural operator**.
>
> 1. **Explicit force field**: Prior methods incorporating continuous time, such as Graph Neural Ordinary Differential Equations (GNN-ODEs), typically use the GNN to learn a latent, **implicit** state transition function, $d z/d t = f_{\text{latent}}(z)$. In contrast, NFF's key innovation is a more explicit representation that enables physics-grounded first-order and second-order integration.
> 2. **Neural operator learning (OOD generalization)**: Traditional function approximators fail when the input composition drastically changes (OOD). NFF uses DeepONet to learn a continuous, dynamic map from object configuration to the resulting force field. This learned operator can **generalize the learned force patterns** (e.g., gravity, collisions) to new object compositions and interaction graphs simply through force summation on the new graph, which is the mechanism that facilitates its superior **cross-scenario generalization**.
>
> The ablation study results strongly support this claim, showing that models lacking the explicit ODE grounding and Neural Operator Learning components (w/o ODE $\rightarrow$ IN; w/o NOL) suffer a massive increase in cross-scenario generalization error compared to NFF.
>
> > [Q1] Could you provide some details on the model update? How many optimization steps are done in each round in Figure 6?
>
> The planning process is an active, closed-loop cycle of mental simulation, physical execution, and model refinement:
> - Mental simulation: The current NFF model (the learned neural simulator) is used to evaluate 500 randomly sampled action candidates in parallel to find the sequence most likely to succeed in the given goal-directed task.
> - Execution: The optimal action sequence is executed in the physical environment (the simulator).
> - Refinement: If the execution **fails** (i.e., the trajectory deviates from the desired goal state), the **observed trajectory (the new data)** is immediately incorporated into the model's optimization via the MSE loss, following the same training procedure used for initial learning. This mechanism allows the model to rapidly adapt its physics understanding based on its own failed experiments.
> We optimize the model for 200 steps in each round. To better present the concept of interaction and planning, we will revise the manuscript and change the term "interactive reasoning" to "interactive planning".

---

> > ### Author Response · Authors · 2025-11-21
> > **Response to reviewer SNun (Part 2)**
> >
> > > [Q2] All of the compared methods are doing pixel reconstructions as far as I can tell.  I wonder how SlotFormer would perform if it’s at least predicting positions (if not forces). Or is it the case for only PHYRE tasks?
> >
> > We appreciate the opportunity to clarify the comparison setup, as maintaining a fair benchmark is crucial.
> > - **I-PHYRE and N-body (State-Based Tasks)**: For these tasks, to ensure a direct comparison of the dynamics model itself, we **decoupled the perceptual component of SlotFormer**. We fed the transformer module directly with the **ground-truth state information** (position, velocity, etc.) and supervised the model in state space. This allowed SlotFormer to focus solely on slot-wise temporal modeling to predict the next object state, making the output directly comparable to our NFF and all other baselines including IN, EGNN, and SEGNO.
> > - **PHYRE (Vision-Based Task)**: For this complex vision-based benchmark, we relied on the original SlotFormer implementation. The full SlotFormer architecture have two components trained seperately: an unsupervised object discovery module (slot attention) and a temporal modeling transformer. We utilized the system's ability to decode the next visual frame based on its predicted latent slots. We do acknowledge a slight unfairness in comparing NFF and RPIN to Slotformer, given that they receives ground-truth object segmentation as input. However, we examined SlotFormer's slot-attention module pretrained on PHYRE, and found it could decouple objects from input images rather well. Therefore, we think that the module is not the primary reason for SlotFormer's poor generalization in cross-scenario settings.

---

> > > ### Comment · Reviewer_SNun · 2025-11-26
> > >
> > > Thank you for your response to my questions.
> > >
> > > **Re: explicit force field**
> > >
> > > Isn't $\frac{dz}{dt} = f_{\text{latent}}(z)$ essentially a force prediction? I'm not sure I follow why that would be implicit. Is it because the integration is over discrete differences?
> > >
> > > Regardless, there are already methods that predict force (or discrete timestep differences) with neural networks, as you already noted in Appendix A, and works that combine them with graph neural nets. So to me, it looks like the main gimmick is the ODE integration, and if there are no works that incorporate that, I think it would be clearer to emphasize it in the related work section.

---

> > > > ### Author Response · Authors · 2025-11-27
> > > >
> > > > We sincerely thank the reviewer for your continued engagement and the positive evaluation of our work, and we appreciate the opportunity to further clarify what we mean by an "explicit force field" and how it differs from previous GNN-ODE style methods.
> > > >
> > > > Whereas GNN-ODEs evolve latent vectors $\frac{dz}{dt}$ in a high-dimensional embedding space (e.g., $z$ is 256D), our method operates directly on interpretable quantities—explicit positions and velocities—whose dynamics are defined in a low-dimensional physical space ($z$ is 2D or 3D). Besides, in implicit methods, aggregation of $f_{\text{latent}}(z)$ occurs entirely in the latent space. This mixes interaction effects and prevents attribution of object-wise contributions after decoding, leading to entangled dynamics. Such entanglement makes few-shot adaptation and OOD transfer challenging, as the latent representation $z$ can drift unpredictably under novel object configurations. In contrast, explicit formulations and the use of neural operators decompose the total force into object-specific components. This preserves the physical structure of interactions and supports better generalization across unseen scenes.
> > > >
> > > > We will further clarify this difference and also emphasize the ODE integration part in related works according to your advice to make it clearer.

---

### Official Review · Reviewer_mX4Q · 2025-11-01

**Soundness:** 2
**Presentation:** 2
**Contribution:** 2
**Rating:** 6
**Confidence:** 4

**Summary:**

This paper proposes a new representation of scene dynamics using neural force fields. The model takes the state of the system as input, represented using position and velocity observations for various objects in the scene as an interaction graph, and produces an estimate of the forces that are operating in the scene. These force estimates can then be used to solve an ODE that would produce future states (object positions and velocities) of the world, conditioned on past observations.

The authors show that this framework enables both prediction and planning tasks with strong performance on multiple synthetic datasets.  The model also is capable of generalizing from much fewer observations compared to existing approaches.

**Strengths:**

1. The methods introduced in this paper are important advances. The force field representation provides clear advantages over the standard interaction network based architectures which are based on latent space transitions. By modeling physical transitions through forces, NFF provides better physical grounding to enable generalization from a lesser number of observations.
2. Neural force fields enable multiple tasks to be performed under the same underlying framework. The authors report strong results on future prediction, forward and backward planning on multiple synthetic datasets like I-Phyre, Phyre and N-body.
3. Experiments in this paper are extensive. Different aspects of the framework such as grounding the ODE and cross scenario generalization are rigorously tested.

**Weaknesses:**

The paper is easy to follow overall, but I found that some aspects of the method are not very clear.

1. The formulation requires the past history of the object states as input in order to predict forces, but it seems like eqn (1) only contains the current state of the world as input. In theory this would not work, unless you make the assumption that acceleration is not required for estimating forces, which seems incorrect. Later in L194, it’s said that the framework uses autoregressive prediction of future states given past states - it seems like this is inconsistent with eqn (1) somehow.
2. I’m wondering how object geometry gets modeled in this framework, because that doesn’t seem to be part of the description of the system state which includes only position and velocity. Isn’t that necessary to predict forces that result from collisions? It seems like there’s some underlying assumptions being made which is not evident when you read the paper.
3. I think another assumption being made is that all collisions are elastic and objects are rigid. To generalize this, we would need a representation of the material properties of objects, which the authors do not discuss. Moreover, it seems like the mass of the object is not part of the state in eqn (1). It might be worth first talking about a more generic physical system which doesn’t have these assumptions, and then introduce the approximations being made. Otherwise it seems like the equations are physically incorrect.
4. The authors talk about an image space NFF, but don’t describe how the state of the system is represented in this case. Do they use an off the shelf model to get object masks/positions, or use pre-trained features as state representation? This part was not clear to me.

The goals laid out in the intro are too broad and ambitious. The paper doesn’t really solve them, which is okay, but the limitations and the way forward should still be discussed (i.e. how might one extend this to real stimuli). Otherwise, it’s hard to understand the contributions of the paper and makes the intro a bit misleading. In fact, the conclusions seem to talk about other domains like migration patterns, etc. which I think is not useful in the light of physical reasoning goals setup in the intro.

**Questions:**

My main questions are about the details of the method, and the assumptions being made on each task introduced in the results section (see point 1-4 in weakness).

I’m also curious to know what the authors think about extending this method to real stimuli and what roadblocks one might face.

---

> ### Author Response · Authors · 2025-11-21
> **Response to reviewer mX4Q**
>
> > [W1] eqn (1) and autoregressive prediction
>
> We would like to clarify that the framework is actually consistent because the state vector $z$ is explicitly defined to contain **both zero-order and first-order physical states** (position, angle, velocity, and angular velocity).
> 1. **Force Dependency**: Since we model forces like gravity, friction, and collision forces, the force function $F(z^q(t))$ directly depends on the zero-order and first-order derivatives. In other words, the state vector $z(t)$ at time $t$ contains sufficient information to predict the next infinitesimal change in state $z(t+dt)$ , which resolves the need for explicitly modeling past history beyond the zero and first derivative at time $t$.
> 2. **Autoregressive Prediction**: The "autoregressive prediction" refers to the iterative simulation process: the ODE solver uses the instantaneous force $F(z^q(t))$ to compute the next state $z(t+dt)$ , and this newly predicted state is then fed back to predict the subsequent step. This process is continuous and locally Markovian, requiring only the comprehensive current state $z(t)$ for the next infinitesimal step.
>
> Therefore, Equation (1) is sufficient for continuous-time modeling without relying on explicit multi-step history.
>
> > [W2] how object geometry gets modeled in this framework
>
> Sorry for the confusion. We agree that the geometry states are necessary for collision detection and forces prediction. Actually, in our implementation, the geometry states have been included in z(t) as input, such as the angle of the cups, the length of blocks, and the radius of the balls in PHYRE and I-PHYRE. Besides, in N-body, the mass of each body is also treated as input in z(t). We will complete these descriptions in L156 to make the formulation more accurate:  "The state vector z incorporates geometry states (angle, length, radius), optionally physical attributes (mass), zero-order states (position), and first-order states (velocity, angular velocity). For image-based NFF, we utilize object masks to replace the geometry states in input."
>
> > [W3] I think another assumption being made is that all collisions are elastic and objects are rigid. It seems like the mass of the object is not part of the state in eqn (1)
>
> We appreciate your suggestions on building a more general formulation of the framework. We are trying to incorporate different materials by including a stress field to model the deformation. Please see the general formulation and our additional results on deformable 3D objects at this [linked pdf](https://anonymous.4open.science/r/ICLR26_Rebuttal-E7D6/formulation.pdf).
>
> We apologize for any possible misunderstanding that may have arisen regarding the mass term. We have revised equations (2) and (4) in the main text to explicitly introduce the mass term into the equations and have added a discussion concerning the mass term, which you may refer to Line 167.
>
> > [W4] how the state of the system is represented in image-based NFF
>
> We obtained the ground-truth segmentation masks for each object from the PHYRE simulator, following the same setting as RPIN. Subsequently, we used CNN as the visual shape encoder. The following procedure is the same as that in the state-based NFF except that the predicted forces are used to transform the object masks, not only the positions. The visual encoder and the dynamics module are optimized end-to-end.
>
> > [Q1] I’m also curious to know what the authors think about extending this method to real stimuli and what roadblocks one might face.
>
> Thank you for your question about extending our method to real-world stimuli. A promising direction is to integrate a 3D reconstruction pipeline—such as Gaussian Splatting—to convert real scenes into an object-centric virtual environment. We can then use the reconstructed 3D points to predict future dynamics and render the results back into video. Building on prior work in this area [1–2], we are eager to explore how to incorporate a Neural Force Field into this framework.
>
> [1] Xie T, Zong Z, Qiu Y, et al. Physgaussian: Physics-integrated 3d gaussians for generative dynamics. CVPR. 2024.
>
> [2] Liu S, Ren Z, Gupta S, et al. Physgen: Rigid-body physics-grounded image-to-video generation. ECCV, 2024.

---

### Official Review · Reviewer_fnPM · 2025-11-03

**Soundness:** 3
**Presentation:** 4
**Contribution:** 3
**Rating:** 6
**Confidence:** 3

**Summary:**

The paper aims to better model and learn the physical interactions, and proposes to use a neural operator to learn the force field from external interventions and object interactions.

The force field is parameterized by a graph neural network. An ODE integrator (such as the Euler integrator) is applied to generate object trajectories from the force field represented by the neural network. To train the model, the network is optimized by minimizing the Mean Squared Error (MSE) loss between predictions and ground truth.

The method is evaluated by several benchmarks: I-PHYRE, a suite of complex physical reasoning puzzles;  N-body problem for trajectory prediction; and PHYRE, a vision-based benchmark of physical reasoning puzzles. Several methods are used to compare against the proposed method: vanilla IN, SEGNO, and SlotFormer. From the experiment results, the proposed method generates trajectories that closely match ground truth behavior, and shows good generalization to new scenario. The trained NFF model also shows the ability to generate plans for new tasks after learning from limited examples.

**Strengths:**

* It appears novel (to me) to learn the force field using a neural operator instead of directly learning physical interactions or trajectories.

* Besides comparing experiments with the ground-truth trajectories, the paper also demonstrates the learned force field, as well as testing it with planning tasks.

* Additionally, the paper provides a clear video to demonstrate the experiment results.

**Weaknesses:**

I just have some additional questions listed in the section below.

**Questions:**

* The paper uses graph based model as the backbone network to model  the force field, would it suffer the issue that hard to model long-distance interactions among multiple objects?


* In traditional physical simulations, force-based method usually needs small time steps to make it stable. Does the same apply to this paper’s force-based method? (Does the model needs a smaller dt to maintain stability, or just need the similar dt as the other compared methods?)


* For the training trajectory that contains multiple time step, is the MSE loss is computed on each time step's predication or only on the final state of the trajectory?


* For the rigid body collision problems tested in the paper, it might be more convincing to also demonstrate such cases in 3D scenes (if an appropriate dataset is available). Most of the examples (excepts for the N-body problem) presented in the paper are 2D cases.

---

> ### Author Response · Authors · 2025-11-21
> **Response to reviewer fnPM**
>
> > [Q1] The paper uses graph based model as the backbone network to model the force field, would it suffer the issue that hard to model long-distance interactions among multiple objects?
>
> We have not observed this problem yet. Due to the use of integration, long-range dependencies can naturally propagate within sub-timesteps. We chose a GNN rather than a transformer as our backbone network because physics in intuitive physics problems is typically sparse, and graph models can adapt well to this characteristic.
>
> > [Q2] In traditional physical simulations, force-based method usually needs small time steps to make it stable. Does the same apply to this paper’s force-based method? (Does the model needs a smaller dt to maintain stability, or just need the similar dt as the other compared methods?)
>
> Yes, our method requires a smaller dt to ensure stability. For a fair comparison, we adopted the same integration dt for NFF and our baseline method SEGNO. However, an advantage of NeuralODE modeling compared to traditional autoregressive prediction is the ability to use methods like adaptive methods during the inference stage to estimate errors and accelerate inference. We show the acceleration achieved using adaptive integration methods on N-Body datasets.
>
> | Model | within | within[0,3T] | cross[0,3T] | Integration steps (relative) ↓ |
> | :--- | :--- | :--- | :--- | :--- |
> | NFF (adaptive tol=1e-3) | 0.120 ± 0.016 | 0.531 ± 0.035 | 1.299 ± 0.028 | 0.105 |
> | NFF (adaptive tol=1e-4) | 0.101 ± 0.012 | 0.491 ± 0.025 | 1.256 ± 0.025 | 0.165 |
> | NFF (adaptive tol=1e-5) | 0.092 ± 0.010 | 0.490 ± 0.025 | 1.251 ± 0.025 | 0.235 |
> | NFF (euler 5e-3) | 0.120 ± 0.011 | 0.557 ± 0.025 | 1.251 ± 0.025 | 1 |
>
> > [Q3] For the training trajectory that contains multiple time step, is the MSE loss is computed on each time step's predication or only on the final state of the trajectory?
>
> It is computed on each time step's prediction.
>
> > [Q4] For the rigid body collision problems tested in the paper, it might be more convincing to also demonstrate such cases in 3D scenes (if an appropriate dataset is available). Most of the examples (excepts for the N-body problem) presented in the paper are 2D cases.
>
> Thank you for your advice. We have added a set of experiments including deformable objects in 3D space to demonstrate the adaptability of our method. We create 3000 3D scenes using MPM in which objects fall and collide with each other. The comparison of prediction results with other baseline models show that our model can better learn the underlying physics and generalize to unseen scenarios. Please refer to the videos of predicted results at this [anonymous link](https://anonymous.4open.science/r/ICLR26_Rebuttal-E7D6).
>
> | Model | Generalization to unseen scenes RMSE ↓ | Generalization to longer timesteps RMSE ↓ |
> | :--- | :--- | :--- |
> | VLM-MPM | 0.306 | 0.328 |
> | GCN | 0.134 | 0.174 |
> | Pointformer | 0.096 | 0.129 |
> | **Our NFF** | **0.082** | **0.107** |

---

### Meta-Review · Area_Chair_2txK · 2026-01-06

**Summary:**

The paper introduces neural force fields, a framework for physical reasoning that learns to map object configurations to explicit continuous force fields. This method uses DeepONet and neural ODEs to model physics from first principles in a cartesian space instead of latent space transitions. This is achieved by the model predicting interpretable physical forces which are then integrated via an ODE solver to produce trajectories.

The key advantage is data efficiency and OOD generalization. The authors demonstrate that by learning the underlying rules of force (gravity, collision, support) rather than just memorizing trajectories, their method can generalize to novel scenarios with significantly fewer training examples than other SOTA methods.

The reviewers initially expressed concerns regarding the methodology and scope of the proposed framework. Key issues were about the model's ability to handle long distance interactions and its dependence on small integration time steps for stability. There was also a call for broader experimental validation, specifically moving beyond 2D rigid bodies to 3D scenes and deformable objects. Additionally reviewers questioned the model's sensitivity to noise in state observations and requested a clearer distinction between NFF’s explicit force fields and the implicit latent-space ODEs found in existing literature.

The authors effectively addressed these concerns during the rebuttal. They provided a suite of 3D deformable object experiments where NFF significantly outperformed baselines. Other important updates included the addition of explicit mass terms and robustness tests showing the model remains stable under 1-5% state noise. Although 5% state noise is still small, we can assume that multi-modal foundationl models would be able to get to human-level object understanding from pixels (nearing such noise thresholds) and then this approach could become broadly applicable.

**Reviewer Concerns:**

Addressed by rebuttal:
- Concerns about the model being limited to 2D rigid-body cases were addressed by the addition of 3,000 3D scenes featuring deformable objects.
- Reviewers questioned the sensitivity to state errors. The authors demonstrated that NFF remains stable under 1–5% state noise, whereas baselines like SEGNO collapsed.
- Initial ambiguity regarding mass was resolved by explicitly incorporating mass terms into the equations and clarifying how NFF implicitly learns mass-normalized forces from visual cues when exact values are unknown.
- The authors clarified the distinction between NFF’s explicit force field and previous implicit latent-space ODEs, explaining how the neural operator enables superior OOD generalization through force summation on new graphs.

Remaining Issues:
- While the vision-based results on PHYRE were interesting, the model’s performance is still inherently bounded by the quality of the visual encoder.
- The framework currently assumes a deterministic environment.

**Reviewer Scores:**

fnPM: Asked for 3D deformable object experiments and the authors provided a compelling set of experiments (which should be added to the final version of the paper).

mX4Q: Their questions regarding object geometry and how autoregressive updates work to make the framework consistent were addressed in the rebuttal.

SNun: authors clarified the explicit force field representation and improved figure clarity.

2R3x: describing the rebuttal as excellent and noting that the noise-robustness and 3D deformable experiments significantly strengthen the paper.

---

### Decision · Program_Chairs · 2026-01-26

Accept (Poster)